# SliceGPT: Compress Large Language Models by Deleting Rows and Columns

**Saleh Ashkboos**[*†‡]
ETH Zurich

**Maximilian L. Croci**[*]
Microsoft Research

**Marcelo Gennari do Nascimento**
Microsoft

**Torsten Hoefler**
ETH Zurich

**James Hensman**
Microsoft Research

## Abstract

Large language models have become the cornerstone of natural language processing, but their use comes with substantial costs in terms of compute and memory resources. Sparsification provides a solution to alleviate these resource constraints, and recent works have shown that trained models can be sparsified post-hoc. Existing sparsification techniques face challenges as they need additional data structures and offer constrained speedup with current hardware. In this paper we present SliceGPT, a new post-training sparsification scheme which replaces each weight matrix with a smaller (dense) matrix, reducing the embedding dimension of the network. Through extensive experimentation we show that SliceGPT can remove up to 25% of the model parameters (including embeddings) for Llama-2 70B, OPT 66B and Phi-2 models while maintaining 99%, 99% and 90% zero-shot task performance of the dense model respectively. Our sliced models run on fewer GPUs and run faster without any additional code optimization: on 24GB consumer GPUs we reduce the total compute for inference on Llama-2 70B to 64% of that of the dense model; on 40GB A100 GPUs we reduce it to 66%. We offer a new insight, computational invariance in transformer networks, which enables SliceGPT and we hope it will inspire and enable future avenues to reduce memory and computation demands for pre-trained models. Code is available at:
https://github.com/microsoft/TransformerCompression .

## 1 Introduction

Large language models (LLMs) are neural networks with billions of parameters, trained on trillions of tokens (Zhao et al., 2023). The cost of training an LLM has caused a shift to re-using pre-trained models for multiple tasks, the *foundation model* paradigm. The size of LLMs makes deploying a pre-trained model an expensive undertaking. Many models require multiple GPUs to be able to compute a prediction, and because the models are autoregressive, multiple forward passes of the neural network are needed to generate text responses. It is therefore of widespread interest to reduce the computational requirements of these models, usually performed via post-training techniques referred to as *model compression*.

A majority of model compression techniques fall into one of four categories: distillation, tensor decomposition (which includes low-rank factorization), pruning and quantization (Hoefler et al., 2021; Gholami et al., 2021; Zhu et al., 2023; Gupta & Agrawal, 2021). In this work we focus on pruning, though we hope that our methodology may influence future work on other areas. Whilst pruning methods have been around for some time, many approaches require recovery fine-tuning (RFT) after pruning to maintain performance, making the overall process an expensive and hard-to-scale task. With SliceGPT we compress large models using a single GPU in just a few hours and maintain competitive performance on generation and downstream tasks even without RFT.

---

[*]Equal contribution

[†]Work completed as an intern at Microsoft.

[‡]{saleh.ashkboos, torsten.hoefler}@inf.ethz.ch    {t-mcroci, marceloge, jameshensman}@microsoft.com

Figure 1: Matrix multiplication of the signal $\mathbf{X}$ and a weight matrix $\mathbf{W}$ under different types of sparsity. **Left**: unstructured sparsity, where some elements of $\mathbf{W}$ are zero, and $\mathbf{X}$ is dense. **Middle**: 2:4 structured sparsity, where each block of four weight matrix entries contains two zeros, and $\mathbf{X}$ is dense. **Right**: SliceGPT, where after introducing transformation $\mathbf{Q}$, all the sparsity is arranged to the bottom rows of $\mathbf{W}$ and the corresponding columns of $\mathbf{X}$ are removed.

Pruning methods work by setting some elements of the weight matrices in an LLM to zero, and (optionally) updating the surrounding elements of the matrix to compensate. The result is a sparse pattern which means that some floating point operations can be skipped in the matrix multiplications required in the forward pass of the neural network. The relative speedup of the operations depends on the level of sparsity and the sparsity pattern: more structured sparsity is associated with more computational gain. In contrast to other pruning methods, SliceGPT prunes away (slices off!) entire rows or columns of the weight matrices. Before slicing, we perform a single transformation of the network which leaves the predictions invariant, but allows the slicing to have only a small effect.

The result is that weight matrices are smaller, and the signals passed between blocks of the neural network are smaller too: we reduce the *embedding dimension* of the neural network.

Figure 1 compares our approach with existing sparsity methods. Our contributions are as follows:

1. We introduce the idea of *computational invariance*: we show that we can apply orthogonal-matrix transformations to each weight matrix in a transformer without changing the model.

2. We use this to edit each block in a transformer architecture, such that we are projecting the signal matrix[1] between blocks onto its own principal components. We remove columns or rows of the transformed weight matrices to reduce the model size. We call the transformation and removal of weights SliceGPT.

3. We conduct multiple experiments on OPT (Zhang et al., 2022) and LLAMA-2 (Touvron et al., 2023) LLMs, demonstrating that SliceGPT is able to compress these models by up to 30% with superior perplexity to the state of the art 2:4 scheme. On downstream tasks we additionally experiment with Phi-2 and show that all models can be sliced by up to 30% while maintaining >90% of the dense performance.

## 2 BACKGROUND

In this section, we first describe some necessary background on transformer architectures, which allows us to introduce notation which we will use to prove our main results. Then we describe related work on sparsification for compressing such architectures.

### 2.1 TRANSFORMER NETWORKS

Transformer networks (Vaswani et al., 2017) are a class of neural networks that have been shown to be effective at a wide range of tasks including language modeling. The transformer architecture is composed of a series of layers, each of which is composed of a multi-head self-attention block followed by a feed-forward network block. Between each block, there is a LayerNorm (Ba et al., 2016) (or RMSNorm (Zhang & Sennrich, 2019)) block. Figure 2 illustrates part of a transformer network: an attention block connected to a Feed Forward Network (FFN) block through a LayerNorm block, with residual connections. The following describes the operations of each component (ignoring dropout, which is not applied post-training).

---

[1] The signal matrix is sometimes referred as activation matrix.

**Embeddings**   Let $D$ be the embedding dimension of our transformer, $N$ be the sequence length. The transformer model takes as input a sequence of token IDs and position IDs, and uses them to index the embedding matrices, producing the initial signal $\mathbf{X}$ with shape $N \times D$. In what follows we consider, without loss of generality, a single embedding matrix $\mathbf{W}_{\text{embd}}$ indexed by input sequence $\boldsymbol{s}$.

**LayerNorm**   After embeddings, the signal matrix is passed through a LayerNorm operation, which subtracts the mean from each row of the matrix, divides the row by its standard deviation, rescales (columnwise), and adds an offset. We write the LayerNorm block as

$$\text{LayerNorm}(\mathbf{X}) = \text{RMSNorm}(\mathbf{XM})\text{diag}(\boldsymbol{\alpha})\sqrt{D} + \mathbf{1}_N\boldsymbol{\beta}^\top \tag{1}$$

where $\text{RMSNorm}(\mathbf{X})$ applies[2] $\boldsymbol{x} \leftarrow \boldsymbol{x}/\|\boldsymbol{x}\|$ to each row of $\mathbf{X}$. The vector parameter $\boldsymbol{\alpha}$ and offset (vector) parameter $\boldsymbol{\beta}$ are learned independently at each LayerNorm instance. The constant matrix $\mathbf{M} = \mathbf{I} - \frac{1}{D}\mathbf{1}\mathbf{1}^\top$ is a $D \times D$ matrix which subtracts the mean from each row of $\mathbf{X}$.

**Attention Blocks**   The attention block has four matrices: $\mathbf{W}_k, \mathbf{W}_q, \mathbf{W}_v$ and $\mathbf{W}_o$, each of dimension $D \times D$. The input signal arriving into the block is projected into the Key ($\mathbf{XW}_k$), Query ($\mathbf{XW}_q$), and Value ($\mathbf{XW}_v$) matrices, which are then split into multiple *heads*. A nonlinear operation is applied at each head before the signals are combined and multiplied by the output weight matrix $\mathbf{W}_o$. Since the first three weight matrices are applied separately to the inputs, we can concatenate them and perform a single matrix multiplication (denoted by the white box around these matrices in Figure 2). We can consider the concatenation of these matrices to be a single linear layer, which we denote $\mathbf{W}_{\text{in}}$. We also refer to the output matrix as $\mathbf{W}_{\text{out}}$. We treat the attention block as $\sigma(\mathbf{XW}_{\text{in}} + \boldsymbol{b}_{\text{in}})\mathbf{W}_{\text{out}} + \boldsymbol{b}_{\text{out}}$[3], where $\sigma$ represents the multi-head attention operation.

**FFN Blocks**   The other type of block that appears in transformer architectures is a Feed Forward Network (FFN) block. In many cases, this is a Multi-layer Perceptron (MLP), which consists of a linear layer $\mathbf{W}_1$, followed by an element-wise operation $\sigma$, followed by a second linear layer: $\sigma(\mathbf{XW}_1 + \boldsymbol{b}_1)\mathbf{W}_2 + \boldsymbol{b}_2$. Some architectures have adopted the gated format, where an additional matrix is used, and the operation is $\big(\sigma(\mathbf{XW}_1 + \boldsymbol{b}_1) \circ (\mathbf{XW}_2)\big)\mathbf{W}_3$, where $\circ$ is an element-wise product. Much like the first three linear layers in the attention module, we can consider the concatenation of $\mathbf{W}_1$ and $\mathbf{W}_2$ to be a single linear operation, and denote it $\mathbf{W}_{\text{in}}$. We can therefore denote the operation of MLP or gated FFN layers as $\sigma(\mathbf{XW}_{\text{in}})\mathbf{W}_{\text{out}}$, where $\sigma$ takes a different meaning to that in an attention.

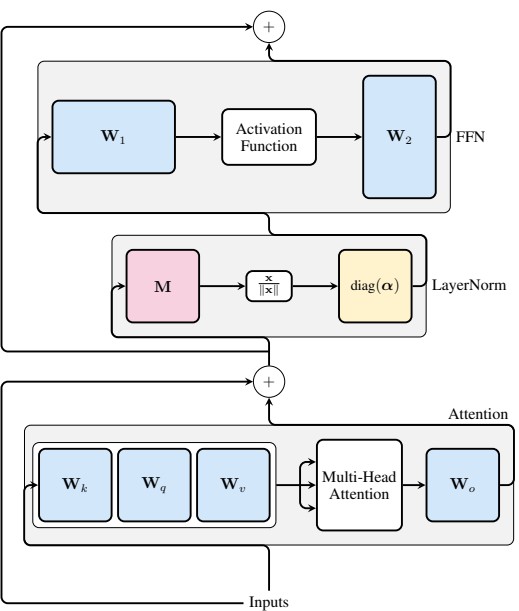

Figure 2:   A single layer in a transformer network. The signals (inputs) arising from the previous blocks of the networks arrive at the bottom of the figure, before being passed through attention, LayerNorm, and FFN. The attention and FFN blocks both have input and output linear operations (blue) which we denote in the text as $\mathbf{W}_{\text{in}}, \mathbf{W}_{\text{out}}$. The linear operations of LayerNorm $\mathbf{M}$ and $\text{diag}(\boldsymbol{\alpha})$ are highlighted. This and subsequent figures do not show biases.

**Language Modelling (LM) Head**   All of the transformer networks to which we apply SliceGPT in this paper have a decoder-only structure following (Radford et al., 2018): after multiple layers applying alternating attention and FFN blocks, a head block computes logits which are used to compute the loss during training and token prediction on deployment. The head operation is $\mathbf{XW}_{\text{head}} + \boldsymbol{b}_{\text{head}}$, where $\mathbf{X}$ is the output of the last transformer block.

---

[2]In some implementations an RMSNorm block may contain scale parameters. We consider these to be special instances of LayerNorm and handle them accordingly.

[3]For ease of notation here and throughout this paper, we abuse notation slightly and omit the broadcasting of the bias terms across the sequence length dimension. The complete notation for the operation of an attention block is $\sigma(\mathbf{XW}_{\text{in}} + \mathbf{1}_N\boldsymbol{b}_{\text{in}}^\top)\mathbf{W}_{\text{out}} + \mathbf{1}_N\boldsymbol{b}_{\text{out}}^\top$.

**Forward pass**   Once the model is trained and all of the parameters are set, the computations required in a transformer network to produce predictions involve passing signal matrices from one block to the next until the head node is reached. Since we are able to define both FFN and attention blocks in the form $\sigma(\mathbf{X}\mathbf{W}_{\text{in}} + \boldsymbol{b}_{\text{in}})\mathbf{W}_{\text{out}} + \boldsymbol{b}_{\text{out}}$, where we understand that $\sigma$ represents either a point-wise or multi-head-attention nonlinearity, we are able to describe the forward pass using Algorithm 1.

---

**Algorithm 1** The forward pass of a transformer network

---

**Require:** $\{\mathbf{W}_{\text{in}}^{\ell}, \boldsymbol{b}_{\text{in}}^{\ell}, \mathbf{W}_{\text{out}}^{\ell}\,\boldsymbol{b}_{\text{out}}^{\ell}\}_{\ell=1}^{L}$               *// weights and biases of FFN and attention blocks*
**Require:** $\{\sigma_{\ell}\}_{\ell=1}^{L}$                        *// nonlinearity associated with each block*
**Require:** $\{\text{Norm}_{\ell}\}_{\ell=0}^{L}$         *// LayerNorm or RMSNorm instances to perform between blocks*
**Require:** $\mathbf{W}_{\text{embd}}, \mathbf{W}_{\text{head}}, \boldsymbol{b}_{\text{head}}$                    *// embedding and head matrices*
**Require:** $\boldsymbol{s}$                              *// input sequence*
1:   $\mathbf{X} \leftarrow \mathbf{W}_{\text{embd}}[\boldsymbol{s}, :]$                 *// index embeddings*
2:   $\mathbf{X} \leftarrow \text{Norm}_0(\mathbf{X})$                *// normalize*
3: **for** $\ell = 1 \dots L$ **do**
4:     $\mathbf{Z} \leftarrow \sigma_{\ell}\big(\mathbf{X}\mathbf{W}_{\text{in}}^{\ell} + \boldsymbol{b}_{\text{in}}^{\ell}\big)\mathbf{W}_{\text{out}}^{\ell} + \boldsymbol{b}_{\text{out}}^{\ell}$       *// apply FFN or attention*
5:     $\mathbf{X} \leftarrow \text{Norm}_{\ell}(\mathbf{X} + \mathbf{Z})$     *// normalize and apply residual connection*
6: **end for**
7: **return** $\mathbf{X}\mathbf{W}_{\text{head}} + \boldsymbol{b}_{\text{head}}$                 *// apply model head*

---

## 2.2   RELATED WORK

In the simplest setting, one can employ magnitude-based sparsification, which involves setting the smallest weights in the model to zero (Han et al., 2016; Zhu & Gupta, 2017; Gale et al., 2019). Although magnitude sparsification is scalable, its application to LLMs gives too strong a degradation in performance (Frantar & Alistarh, 2023). Optimal Brain Surgeon (OBS) (Hassibi et al., 1993; LeCun et al., 1989), a more sophisticated method, systematically removes weights that have the least impact on the loss function. The method compensates for the error introduced by weight removal by updating the un-pruned weights using the inverse of the Hessian matrix. Unfortunately, OBS is impractical for models with a few million parameters due to the need to calculate and store the inverse of the Hessian matrix. To address the computational limitation posed by OBS, recent research has explored two approaches: approximating the inverse of the Hessian matrix such as WoodFisher (Singh & Alistarh, 2020) or applying it separately to each layer such as in Optimal Brain Compression (OBC, Frantar & Alistarh, 2022), known as layer-wise pruning. While these techniques have proven effective for medium-sized networks, they are not practical for large language models, where individual layer weight matrices typically contain more than $10^8$ parameters.

GPTQ (Frantar et al., 2022) has solved this issue by quantizing (representing the parameter using lower precision) the weight matrix of LLMs using a column-by-column scheme and updating all not-yet-quantized weights in the next columns. SparseGPT (Frantar & Alistarh, 2023) applied the same idea for pruning and sparsifies the LLMs using unstructured and semi-structured pruning, and Sun et al. (2023) simplified the idea by using only the diagonal of the Hessian. Since achieving end-to-end speed improvements through unstructured pruning is a demanding task, they also attempted a similar technique to induce sparsity with semi-structured patterns like 2:4 and 4:8 (Mishra et al., 2021). However, implementing such structures does not maintain the accuracy of the model.

Another approach to compression is low-rank approximation, where each weight matrix is replaced with the product of two matrices with a smaller inner dimension, usually followed by a fine-tuning step (Hu et al., 2021; Mahabadi et al., 2021; Noach & Goldberg, 2020; Tukan et al., 2020). To achieve compression, the inner dimension must be smaller than half of the original dimension. In contrast, our method replaces each weight matrix with a single smaller one, reducing the embedding dimension without the need for fine-tuning.

We propose to delete rows and columns of weight matrices, which is similar to pruning of filters and channels in the convnet literature. There, sparsity-inducing regularization is added to batch-norm factors (Liu et al., 2017) or network structures (Huang & Wang, 2018), and the network is trained or fine-tuned, resulting in the pruning of channels or parts of the network. Perhaps the most analogous methods to ours are ThiNet (Luo et al., 2017; He et al., 2017), which apply linear operations between layers (as will we), interleaved with more fine-tuning with regularization. In this literature, the model sizes are typically several orders of magnitude smaller than in LLMs, for example the VGG16 network has 138M parameters, comparable with the very smallest OPT model that we consider. The

huge size of LLMs makes methods that involve extensive fine-tuning unappealing, especially when outer-loops are needed to select regularization parameters.

Recently, some works have been proposed that apply structured pruning to LLMs, followed by continued training (or fine-tuning) to recover the performance that is lost. For example LLM-pruner (Ma et al., 2023a) removes connected structures from an LLM before further training. Contemporarily with our work, LLM Surgeon (van der Ouderaa et al., 2023) interweaves recovery fine-tuning with pruning. We provide results for SliceGPT as a single-shot method and with post-slicing recovery fine-tuning.

## 3 SLICEGPT

Our SliceGPT method relies on a computational invariance that is inherent in the transformer architecture. By this, we mean that it is possible to apply an orthogonal transformation to the output of one component, so long as it is undone in the next. Our key insight is that the RMSNorm operation which is performed between blocks of the network does not affect the transformation: the operations commute. In this section, we first describe how the invariance occurs in RMSNorm-connected transformer networks, then we note how networks trained with LayerNorm connections can be converted to RMSNorm. Next, we describe our method to compute transformations at each layer using Principal Component Analysis (PCA), such that the signal between blocks is projected onto its principal components. Finally, we describe how deleting the minor principal components corresponds to slicing away rows or columns of the modified network.

### 3.1 COMPUTATIONAL INVARIANCE IN TRANSFORMER NETWORKS

Let $\mathbf{Q}$ denote an orthogonal matrix: we have $\mathbf{Q}^\top \mathbf{Q} = \mathbf{Q}\mathbf{Q}^\top = \mathbf{I}$. Note that multiplying a vector $\boldsymbol{x}$ by $\mathbf{Q}$ does not change the norm of the vector, since $\|\mathbf{Q}\boldsymbol{x}\| = \sqrt{\boldsymbol{x}^\top \mathbf{Q}^\top \mathbf{Q}\boldsymbol{x}} = \sqrt{\boldsymbol{x}^\top \boldsymbol{x}} = \|\boldsymbol{x}\|$. In this work, the dimensions of $\mathbf{Q}$ will always match the embedding dimension of the transformer $D$.

Suppose that $\mathbf{X}_\ell$ is the output of one block of the transformer, which is then processed by RMSNorm, and then inputted to the subsequent block as RMSNorm($\mathbf{X}_\ell$). If we insert linear layers with the orthogonal matrix $\mathbf{Q}$ before RMSNorm and $\mathbf{Q}^\top$ after RMSNorm, the network remains unchanged, since each row of the signal matrix is multiplied by $\mathbf{Q}$, normalized and multiplied by $\mathbf{Q}^\top$. We have

$$\text{RMSNorm}(\mathbf{X}_\ell \mathbf{Q})\mathbf{Q}^\top = \text{RMSNorm}(\mathbf{X}_\ell). \tag{2}$$

A proof of this relation appears in Appendix A.1. Now, since each attention or FFN block of the network has a linear operation on both the input and output, we can absorb the additional operations $\mathbf{Q}$ into the linear layers of the blocks. Since the network contains residual connections, we must also apply $\mathbf{Q}$ to the output of all previous layers (all the way back to the embedding) and to all subsequent layers (all the way up to the LM Head).

An *invariant* function is one for which a transformation to the input does not result in a change to the output. In our case, we can apply any orthogonal transformation $\mathbf{Q}$ to the weights of the transformer without changing the result, so the *computation* can be performed in any transformed state. We refer to this as a *computational invariance*, and define it in the following theorem.

**Theorem 1.** *Let $\mathbf{W}_{in}^\ell$ and $\mathbf{W}_{out}^\ell$ be the weight matrices of the linear layers of the $\ell$-th block of an RMSNorm-connected transformer network, and $\boldsymbol{b}_{in}^\ell, \boldsymbol{b}_{out}^\ell$ be the corresponding biases, if any, and let $\mathbf{W}_{embd}$ and $\mathbf{W}_{head}$ be the embedding and head matrices. Let $\mathbf{Q}$ be an orthogonal matrix of dimension $D$. Then the following network is equivalent to the original transformer network:*

$$\tilde{\mathbf{W}}_{embd} = \mathbf{W}_{embd}\mathbf{Q}, \tag{3} \qquad\qquad \tilde{\boldsymbol{b}}_{out}^\ell = \mathbf{Q}^\top \boldsymbol{b}_{out}^\ell, \tag{6}$$

$$\tilde{\mathbf{W}}_{in}^\ell = \mathbf{Q}^\top \mathbf{W}_{in}^\ell, \tag{4} \qquad\qquad \tilde{\mathbf{W}}_{head} = \mathbf{Q}^\top \mathbf{W}_{head}. \tag{7}$$

$$\tilde{\mathbf{W}}_{out}^\ell = \mathbf{W}_{out}^\ell \mathbf{Q}, \tag{5}$$

*The input and head biases are copied: $\tilde{\boldsymbol{b}}_{in}^\ell = \boldsymbol{b}_{in}^\ell$, $\tilde{\boldsymbol{b}}_{head} = \boldsymbol{b}_{head}$.*

*Proof.* We can show that the transformed network computes the same results as the original by stepping through Algorithm 1. Suppose that on line 1, the original network has computed $\mathbf{X}$, then

the modified network has computed $\tilde{\mathbf{X}} = \mathbf{X}\mathbf{Q}$, using Equation 3. Applying RMSNorm on line 2, we see that the operation of the two networks matches: by Equation 2 we have $\text{RMSNorm}(\tilde{\mathbf{X}}) = \text{RMSNorm}(\mathbf{X}\mathbf{Q}) = \text{RMSNorm}(\mathbf{X})\mathbf{Q}$. Applying the nonlinearity on line 4, we see that $\tilde{\mathbf{X}}\tilde{\mathbf{W}}_{\text{in}}^{\ell} = \mathbf{X}\mathbf{W}_{\text{in}}^{\ell}$, using Equation 4 and it follows that $\tilde{\mathbf{Z}} = \mathbf{Z}\mathbf{Q}$. On line 5 the residual connection means we have $(\tilde{\mathbf{X}} + \tilde{\mathbf{Z}}) = (\mathbf{X} + \mathbf{Z})\mathbf{Q}$, and applying RMSNorm results in assignment of $\tilde{\mathbf{X}} = \mathbf{X}\mathbf{Q}$. This follows through to the end of the loop. Finally, on line 7, the transformations are undone as $\mathbf{X}\mathbf{W}_{\text{head}} = \tilde{\mathbf{X}}\tilde{\mathbf{W}}_{\text{head}}$ using Equation 7. □

## 3.2 LAYERNORM TRANSFORMERS CAN BE CONVERTED TO RMSNORM

The computational invariance of the transformer network applies only to RMSNorm-connected networks. Before working on those with Layer-Norm, we convert the network to RMSNorm by absorbing the linear blocks of LayerNorm into the adjacent blocks. Figure 3 shows such a transformation on the transformer network (see Figure 2) . In each block, we multiply the output matrix $\mathbf{W}_{\text{out}}$ by the mean-subtraction matrix $\mathbf{M}$, which accounts for the mean subtraction that would happen in the subsequent LayerNorm. The input matrices $\mathbf{W}_{\text{in}}$ are pre-multiplied by the scales of the preceding LayerNorm blocks. The embedding matrix $\mathbf{W}_{\text{embd}}$ must be mean-subtracted, and $\mathbf{W}_{\text{head}}$ must be re-scaled by the last Layer-Norm scales. This is a straightforward change in the order of operations and does not affect the network output.

Figure 3: Converting a transformer network from LayerNorm to RMSNorm: the scale matrix $\text{diag}(\boldsymbol{\alpha})$ is absorbed into the subsequent matrix $\mathbf{W}_{\text{in}}$. Figure shows the block in combined colors. We use $(\boldsymbol{\alpha})$ for brevity. The mean-subtraction matrix $\mathbf{M}$ is applied to each matrix $\mathbf{W}_{\text{out}}$. Layer-norm becomes RMSNorm, up to a constant $\sqrt{D}$ (not shown). Here, the scaling $(\boldsymbol{\alpha}')$ comes from the previous block.

## 3.3 A TRANSFORMATION PER BLOCK

Now that every LayerNorm in the transformer has been converted to RMSNorm, we can select any $\mathbf{Q}$ to modify the model. Our initial plan was to collect signals from the model, construct an orthogonal matrix using those signals and to delete parts of the network. We quickly saw that the signals at different blocks of the network were not aligned, and that we would need to apply a different orthogonal matrix at each block, $\mathbf{Q}_{\ell}$.

Allowing the orthogonal matrix used in each block to differ can be shown to leave the model unchanged using the same proof as Theorem 1, with the exception of line 5 of Algorithm 1. Here we see that the residual connection and the output of the block must have the same rotation. To fix this, we modify the residual connection by applying the linear transformation $\mathbf{Q}_{\ell-1}^{\top}\mathbf{Q}_{\ell}$ to the residual. Figure 4 shows how different rotations can be applied to different blocks with the additional linear operation in the residual connection. Unlike the modifications to the weight matrices, these additional operations cannot be pre-computed and add a small ($D \times D$) overhead to the model. Nonetheless, they are needed to allow slicing the model (Section 3.4) and we see real speedup overall (Section 4).

To compute the matrices $\mathbf{Q}_{\ell}$, we use PCA. We select a calibration dataset from the training set, run it through the model (after converting LayerNorm operations into RMSNorm), and extract the orthogonal matrix of the layer. We use the output of the transformed network to calculate the orthogonal matrices of the next layers. More precisely, if $\mathbf{X}_{\ell,i}$ is the output of the $\ell^{\text{th}}$ RMSNorm block for the $i^{\text{th}}$ sequence in the calibration dataset, we compute

$$\mathbf{C}_{\ell} = \sum_{i} \mathbf{X}_{\ell,i}^{\top}\mathbf{X}_{\ell,i} \tag{8}$$

and set $\mathbf{Q}_{\ell}$ to the be the eigenvectors of $\mathbf{C}_{\ell}$, sorted by decreasing eigenvalues.

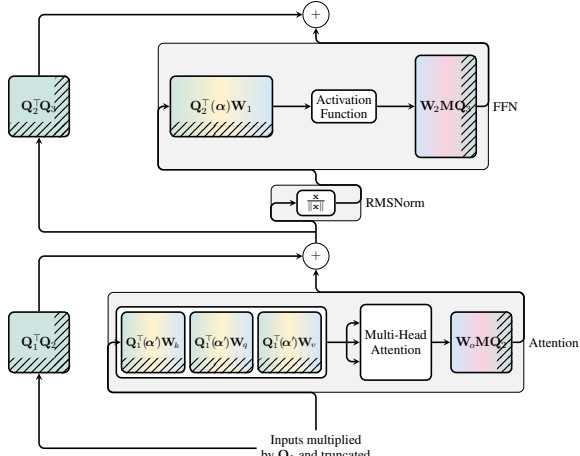

Figure 4: With the network converted to RMSNorm (see Figure 3), we apply the computational-invariance idea. The input weight matrices $\mathrm{diag}(\boldsymbol{\alpha})\mathbf{W}_{\mathrm{in}}$ are pre-multiplied by $\mathbf{Q}^\top$. The output matrices $\mathbf{W}_{\mathrm{out}}\mathbf{M}$ are post-multiplied by $\mathbf{Q}$. In the skip-connection, a new linear layer is added $\mathbf{Q}_\ell^\top\mathbf{Q}_{\ell+1}$. After these modifications, the matrices can be sliced (hatched areas).

## 3.4 SLICING

The goal of Principal Component Analysis is usually to take a data matrix $\mathbf{X}$ and compute a lower dimensional representation $\mathbf{Z}$, and an approximate reconstruction $\tilde{\mathbf{X}}$:

$$\mathbf{Z} = \mathbf{XQD}, \qquad \tilde{\mathbf{X}} = \mathbf{ZD}^\top\mathbf{Q}^\top. \qquad (9)$$

where $\mathbf{Q}$ is the eigenvectors of $\mathbf{X}^\top\mathbf{X}$, and $\mathbf{D}$ is a $D \times D_{\mathrm{small}}$ deletion matrix (containing $D_{\mathrm{small}}$ columns of the $D \times D$ identity matrix), which removes some of the columns of the matrix to the left. The reconstruction is $L_2$ optimal, in the sense that $\mathbf{QD}$ is a linear mapping that minimizes $\|\mathbf{X} - \tilde{\mathbf{X}}\|^2$.

When we apply PCA to the signal matrix $\mathbf{X}$ between blocks, we never materialize the $N \times D$ signal matrix, but we apply the deletion matrix $\mathbf{D}$ to the operations preceding and succeeding the construction of that matrix, which have already been multiplied by $\mathbf{Q}$ in the above. We delete rows of $\mathbf{W}_{\mathrm{in}}$ and columns of $\mathbf{W}_{\mathrm{out}}$ and $\mathbf{W}_{\mathrm{embd}}$. We also delete both rows *and* columns of the matrix $\mathbf{Q}_{\ell-1}^\top\mathbf{Q}_\ell$ that we have inserted into the residual connection (see Figure 4).

## 4 EXPERIMENTAL VALIDATION

**Setup**  We use Hugging Face Transformers (Wolf et al., 2019) to implement our code with PyTorch (Paszke et al., 2019). The computation of $\mathbf{Q}$ is performed on a single H100 GPU with 80GB of memory, taking approximately 3.5 hours to complete for the LLAMA-2 70B model. We use double precision for the PCA calculation because using single precision for eigenvector calculations in PyTorch leads to a discrepancy in the final accuracy, as detailed in Appendix A.2.

We experiment with two different calibration sets: the WikiText-2 training dataset (Merity et al., 2016) and the Alpaca training dataset (Taori et al., 2023). An ablation study on the calibration set size and sequence length is presented in Appendix A.3. We apply a small amount of RFT to sliced LLAMA-2 and Phi-2 models using LoRA (Hu et al., 2021), following the idea from Ma et al. (2023a). For models sliced with WikiText-2 we use approximately 1k sequences, for those sliced with the Alpaca dataset we use 5k. We use LoRA with $r = 32$, $\alpha = 10$ and sequence length 1024, and defaults for all other hyperparameters in PEFT (Mangrulkar et al., 2022).

**Models, Tasks, and GPUs**  We evaluate all our experiments on OPT (Zhang et al., 2022), LLAMA-2 (Touvron et al., 2023) model families, and additionally evaluate Phi-2 (in our zero-shot task) experiments. We exclude OPT 175B, as it is outperformed by smaller LLAMA-2 models. Nonetheless, we anticipate that this larger model will yield improved results, as larger models typically offer more promising opportunities for compression (see Section 4.1). We evaluate our scheme on both language generation as well as popular zero-shot tasks. To demonstrate the comprehensive speedup achieved by SliceGPT we use: Quadro RTX6000 GPUs with 24GB of memory as a representative example of consumer-level GPUs; 40GB A100s and 80GB H100s to provide datacenter-level benchmarks.

**Baseline Setup**  We initially planned to compare our results against a scheme that pruned columns (or rows) with the smallest norm but found that this baseline was very poor, with the WikiText-2

perplexity of the model soaring into the 1000s after pruning just a few columns. Instead, we compare SliceGPT against SparseGPT (Frantar & Alistarh, 2023) employing a 2:4 sparsity ratio, as this is the only sparsity scheme which achieves speedup (Mishra et al., 2021).

## 4.1 RESULTS

**Generation Task** We begin by showcasing our findings using the WikiText-2 dataset. In this context, we evaluate the performance of both the OPT and LLAMA-2 model families across different sizes when using this dataset for slicing. Table 1 shows the perplexity obtained by various slicing levels. SliceGPT exhibits superior performance when applied to OPT models compared to LLAMA-2 models which matches our intuition from the spectrum analysis of those models (see Appendix A.4 for our discussion). The performance of SliceGPT improves as the model size increases. Comparing SliceGPT with SparseGPT, we see that that SparseGPT 2:4 performs worse than SliceGPT with 25% slicing in all LLAMA-2 models. For OPT, we see that 30% sliced models beat 2:4 sparsity for all model sizes except 2.7B.

Table 1: OPT and LLAMA-2 perplexity results on WikiText2. The calibration set size and sequence length are 1024 and 2048, respectively.

| Method | OPT | | | | | | | LLAMA-2 | | |
|---|---|---|---|---|---|---|---|---|---|---|
| | 125M | 1.3B | 2.7B | 6.7B | 13B | 30B | 66B | 7B | 13B | 70B |
| Dense | 27.64 | 14.61 | 12.46 | 10.85 | 10.12 | 9.56 | 9.33 | 5.47 | 4.88 | 3.32 |
| SparseGPT 2:4 | 45.07 | 29.61 | 14.90 | 13.00 | 11.80 | 10.53 | 10.22 | 8.69 | 7.07 | 4.98 |
| SliceGPT (10%) | 29.34 | 15.10 | 12.75 | 10.92 | 10.27 | 9.65 | 9.43 | 5.89 | 5.21 | 3.69 |
| SliceGPT (20%) | 34.26 | 16.43 | 13.73 | 11.48 | 10.66 | 9.87 | 9.57 | 6.64 | 5.81 | 4.25 |
| SliceGPT (25%) | 37.74 | 17.46 | 14.56 | 11.90 | 10.94 | 10.04 | 9.68 | 7.24 | 6.30 | 4.60 |
| SliceGPT (30%) | 43.98 | 19.09 | 15.83 | 12.51 | 11.33 | 10.27 | 9.85 | 8.12 | 6.99 | 5.05 |

**Zero-shot Tasks** We assess SliceGPT across five well-known zero-shot tasks: PIQA (Bisk et al., 2020); WinoGrande (Sakaguchi et al., 2021); HellaSwag (Zellers et al., 2019); ARC-e and ARC-c (Clark et al., 2018) using the LM Evaluation Harness (Gao et al., 2021). Figure 5 shows the results. We see a marked difference between the datasets, with the Alpaca dataset giving much higher performing models. We attribute this difference to the similarity between Alpaca and the benchmark tasks. For LLAMA-2 70B sliced at 30%, with RFT on Alpaca we are able to achieve an average accuracy of 74.3%, compared to 76.6% on the dense model. The sliced model has approximately 51.6B parameters and considerably improved throughput as we demonstrate later. Results for OPT and for all models post-pruning without RFT are shown in Appendix A.5.

We see that Phi-2 is not able to recover the drop in accuracy from slicing using only the WikiText-2 dataset, but using Alpaca we are able to recover several percentage points. The average accuracy of Phi-2 with 25% slicing and RFT is 65.2%, compared to 72.2% with the dense model. The sliced model has approximately 2.2B parameters and retains 90.3% of the accuracy of the 2.8B model. This shows that even small LMs can benefit from post-training pruning. Tables of accuracies across each task are provided in Appendix A.6.

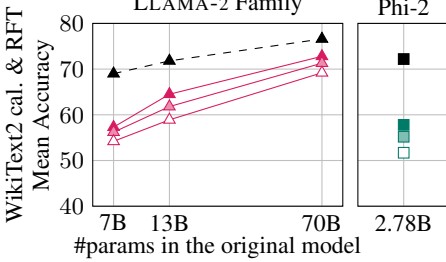 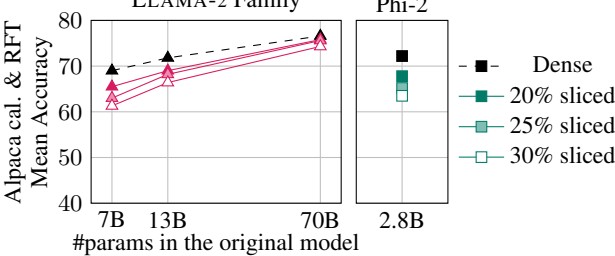

Figure 5: Mean zero-shot accuracy on LLAMA-2 and Phi-2 across multiple tasks after slicing and recovery fine-tuning (RFT). Left: WikiText-2 used for calibration and RFT. Right: Alpaca used for calibration and RFT.

**Benchmarking Throughput** Unlike conventional sparsity methods, which introduce sparsity in $\mathbf{W}_{\text{in}}$ and $\mathbf{W}_{\text{out}}$, SliceGPT also introduces (structured) sparsity in $\mathbf{X}$: entire columns of $\mathbf{X}$ are sliced off, reducing the embedding dimension. This enhances both the computational complexity (in flops) and data movement within our compressed model.

The token throughput of models sliced at 25% and 50% are compared to the dense model on 80GB H100 GPUs. We set the sequence length to 128 and find the maximum throughput by doubling the batch size until the GPUs run out of memory or the throughput drops off. The 25% sliced models achieve up to $1.55\times$ throughput improvement over the dense model. At 50% slicing the largest models require only one GPU instead of two, with large increases in throughput: $3.13\times$ and $1.87\times$. This means that for a fixed number of GPUs, these models achieve $6.26\times$ and $3.75\times$ throughput of a dense model. We note that the WikiText2 perplexity of SliceGPT at 50% is worse than SparseGPT 2:4, but the throughput is much higher than could be achieved with a sparse method that does not slice $\mathbf{X}$. For full details see Appendix A.7.

**Inference Time** Table 2 compares the time of generating a single token in OPT 66B and LLAMA-2 70B models on Quadro RTX6000 and A100 GPUs. We observe 16-17% speedup on RTX6000 GPUs when employing 25% slicing, and 11-13% on A100s. We reduce the number of GPUs used in both cases, providing energy and cost savings relative to deployment of the dense model. For LLAMA-2 70B, the compute required using RTX6000 GPUs is reduced to 64%, from 1764 GPUms to 1075 GPUms[4]. We attribute this improvement to our approach of substituting weight matrices with smaller ones in our compressed models, which is infeasible with other pruning schemes.

Table 2: Average per-token inference time of SliceGPT when generating sequences of length 128 (with batch size of 1). In each case, we show the time taken in ms, the number of GPUs required and the total compute in GPUms.

| GPU Type | Slicing | OPT 66B | | LLAMA-2 70B | |
|---|---|---|---|---|---|
| A100 (40GB) | Dense | 114ms on 4 GPUs | 456 GPUms | 125ms on 4 GPUs | 500 GPUms |
| | 25% | 102ms on 3 GPUs | 306 GPUms | 110ms on 3 GPUs | 330 GPUms |
| Quadro RTX6000 (24GB) | Dense | 237ms on 6 GPUs | 1422 GPUms | 252ms on 7 GPUs | 1764 GPUms |
| | 25% | 204ms on 5 GPUs | 1020 GPUms | 215ms on 5 GPUs | 1075 GPUms |

End-to-end performance gains are not feasible with the SparseGPT baseline at the time of writing. Instead, we compare SliceGPT with SparseGPT by comparing the relative timing of each operation involved in a transformer layer. We find that SliceGPT (25%) is competitive with SparseGPT (2:4) in terms of speedup and perplexity for large models. For further details see Appendix A.8.

## 5 CONCLUSION AND FUTURE WORK

We've introduced SliceGPT which allows for structured pruning for large language models. We reduce the cost of inference of LLAMA-2 70B on 40GB A100 GPUs to 66% of that of the dense model without any additional code optimization, requiring fewer GPUs (from 4 to 3) while maintaining better held-out perplexity than SparseGPT 2:4. On 24GB RTX6000 GPUs, the cost of inference is reduced to 64%, requiring 2 fewer GPUs (from 7 to 5). On zero-shot downstream tasks, slicing OPT 66B, LLAMA-2 70B and Phi-2 at 25% maintains 99%, 96% and 87% of the dense performance. With recovery fine-tuning 25%-sliced LLAMA-2 70B and Phi-2 increase to 99% and 90% respectively.

Opportunities remain to build on our method. Smaller but dense LMs perform better than LMs with 13B parameters or less pruned to similar sizes, though we do not expect this to remain the case for long. Our pruned models have more parameters than those pruned with SparseGPT but fit larger batches in GPU memory with no overhead for sparsity structure: perhaps a combined method could obtain the best of both. Other methods of computing $\mathbf{Q}$ could improve the results. To further decrease the inference time and GPU count, complementary methods including quantization (Xiao et al., 2023; Dettmers et al., 2022; Ashkboos et al., 2023; Dettmers et al., 2023; Frantar et al., 2022), and structural pruning (e.g. Ma et al., 2023b) could be used.

---

[4]Our Hugging Face-based testing does not enjoy continuous batching or model sharding. This means that in terms of inference time, the dense-model could be improved more than our sliced model in terms of GPUms. Nonetheless, our measurements *do* reflect the energy-usage per token in such a deployment.

ACKNOWLEDGEMENTS

We thank Dmitry Kats, Pashmina Cameron, Pavel Myshkov, Elena Pochernina and Liana Mikaelyan for their invaluable contributions to the source code. We additionally thank Pashmina Cameron for her helpful feedback when reviewing early versions of the paper.

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

# A    APPENDIX

## A.1    PROOF OF EQUATION 2

An orthogonal matrix $\mathbf{Q}$ is a square matrix that satisfies the relation $\mathbf{Q}^\top\mathbf{Q} = \mathbf{Q}\mathbf{Q}^\top = \mathbf{I}$. The norm of a vector is the square-root of the sum of squares of the elements: $\|\boldsymbol{x}\| = \sqrt{\sum_i \boldsymbol{x}_i^2} = \sqrt{\boldsymbol{x}^\top\boldsymbol{x}}$. Multiplying a vector by $\mathbf{Q}$ does not change the norm since $\|\mathbf{Q}\boldsymbol{x}\| = \sqrt{\boldsymbol{x}^\top\mathbf{Q}^\top\mathbf{Q}\boldsymbol{x}} = \|\boldsymbol{x}\|$.

The RMSNorm operation divides each row of the input matrix $\mathbf{X}$ by its norm. By the basic rules of linear algebra, if $\boldsymbol{x}$ is a row of $\mathbf{X}$, then $\mathbf{Q}^\top\boldsymbol{x}$ is the corresponding row of $\mathbf{XQ}$. Applying RMSNorm to $\mathbf{XQ}$, said row will now be equal to $\frac{1}{\|\boldsymbol{x}\|}\mathbf{Q}^\top\boldsymbol{x}$. After RMSnorm, we can multiply by $\mathbf{Q}^\top$, our row is now equal to $\frac{1}{\|\boldsymbol{x}\|}\mathbf{Q}\mathbf{Q}^\top\boldsymbol{x} = \frac{1}{\|\boldsymbol{x}\|}\boldsymbol{x}$. Thus we have the relation

$$\text{RMSNorm}(\mathbf{XQ})\mathbf{Q}^\top = \text{RMSNorm}(\mathbf{X})\,. \tag{10}$$

## A.2    SINGLE PRECISION EIGENVALUE CALCULATION

As previously noted in Section 4, we employ double precision when performing the PCA algorithm. This choice is made in order to mitigate potential numerical errors that may arise during the computation of the orthogonal matrix in SliceGPT. Nevertheless, it is intriguing to investigate the impact of employing lower precision for PCA calculations on the ultimate accuracy.

Table 3 shows the perplexity of all our models when we apply FP32 PCA in our scheme. It shows that the accuracy of larger models could be affected by numerical errors during the PCA calculation phase. It should be noted that we use PyTorch (`torch.linalg`) for calculating the eigenvectors and eigenvalues.

Table 3: OPT and LLAMA-2 perplexity results on WikiText2 using FP32 PCA calculation. The calibration set size and sequence length are 128 and 2048, respectively.

| Method | OPT | | | | | | | LLAMA-2 | | |
|---|---|---|---|---|---|---|---|---|---|---|
| | 125M | 1.3B | 2.7B | 6.7B | 13B | 30B | 66B | 7B | 13B | 70B |
| Dense | 27.64 | 14.61 | 12.46 | 10.85 | 10.12 | 9.56 | 9.33 | 5.47 | 4.88 | 3.32 |
| SparseGPT 2:4 | 45.07 | 29.61 | 14.90 | 13.00 | 11.80 | 10.53 | 10.22 | 8.69 | 7.07 | 4.98 |
| SliceGPT 10% | 29.48 | 15.15 | 12.83 | 11.05 | 10.28 | 9.68 | 9.45 | 6.51 | 5.64 | 4.20 |
| SliceGPT 20% | 34.12 | 16.51 | 13.87 | 11.64 | 10.73 | 9.94 | 9.80 | 7.30 | 6.07 | 5.82 |
| SliceGPT 25% | 38.25 | 17.67 | 14.78 | 12.14 | 11.08 | 10.15 | 9.81 | 8.52 | 6.65 | 7.01 |
| SliceGPT 30% | 44.17 | 19.33 | 16.20 | 12.82 | 11.53 | 10.43 | 9.99 | 10.41 | 7.49 | 8.75 |

## A.3    SENSITIVITY TO THE CALIBRATION SET SIZE AND SEQUENCE LENGTH

We present an ablation study to examine the role of the WikiText-2 calibration set. We focus on the generation task with 25% sparsity using OPT 6.7B and LLAMA-2 7B models.

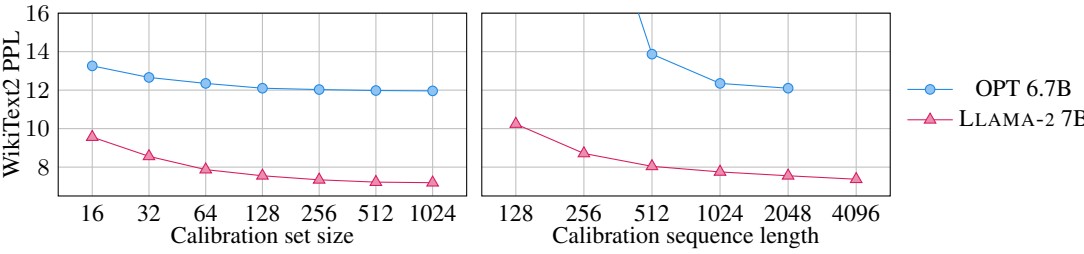

Figure 6: The effect of the calibration set size and sequence length on perplexity of WikiText2.

Figure 6 (left) shows the result of varying the size of the calibration set on the perplexity. It shows that sample sizes of at least 128 provide sensible choices for our calibration set.

Next we explore the effect of using different sequence lengths $N$ in the calibration set. Given a fixed number of $B$ samples, the PCA input matrix is computed using $NB$ embedding vectors, and understanding the tradeoff between having a larger $B$ or a larger $N$ is interesting. Figure 6 (right) shows the results of varying the sequence length in the calibration set from 128 to 4096: we conclude that having a larger sequence length can result in better perplexity.

Using these insights, we use a calibration set size of 1024 and sequence length 2048 in our main experiments (Table 1). In Table 4 below we evaluate the perplexity of OPT and LLAMA-2 models on WikiText-2 with a smaller calibration set size, which confirms the trend that decreasing this degrades the perplexity across all models and sizes.

Table 4: OPT and LLAMA-2 perplexity results on WikiText2. The calibration set size and sequence length are 128 and 2048, respectively.

| Method | OPT | | | | | | | LLAMA-2 | | |
|---|---|---|---|---|---|---|---|---|---|---|
| | 125M | 1.3B | 2.7B | 6.7B | 13B | 30B | 66B | 7B | 13B | 70B |
| Dense | 27.64 | 14.61 | 12.46 | 10.85 | 10.12 | 9.56 | 9.33 | 5.47 | 4.88 | 3.32 |
| SparseGPT 2:4 | 45.07 | 29.61 | 14.90 | 13.00 | 11.80 | 10.53 | 10.22 | 8.69 | 7.07 | 4.98 |
| SliceGPT (10%) | 29.33 | 15.15 | 12.82 | 11.00 | 10.30 | 9.66 | 9.43 | 5.96 | 5.29 | 3.78 |
| SliceGPT (20%) | 34.53 | 16.58 | 13.89 | 11.62 | 10.75 | 9.91 | 9.61 | 6.86 | 6.04 | 4.46 |
| SliceGPT (25%) | 38.13 | 17.78 | 14.84 | 12.12 | 11.08 | 10.10 | 9.76 | 7.56 | 6.61 | 4.89 |
| SliceGPT (30%) | 44.61 | 19.61 | 16.30 | 12.81 | 11.55 | 10.32 | 9.95 | 8.64 | 7.44 | 5.42 |

## A.4 SPECTRUM ANALYSIS OF LLAMA-2 AND OPT MODELS

The figure below shows the eigenvalue distribution for the OPT 6.7B and LLAMA-2 7B models. Although both models have a comparable parameter count, the LLAMA-2 model has a more tightly compressed distribution in its embeddings spectrum. This observation shows that there are no dominant principal components with significantly more information, making the pruning of these components a more challenging task.

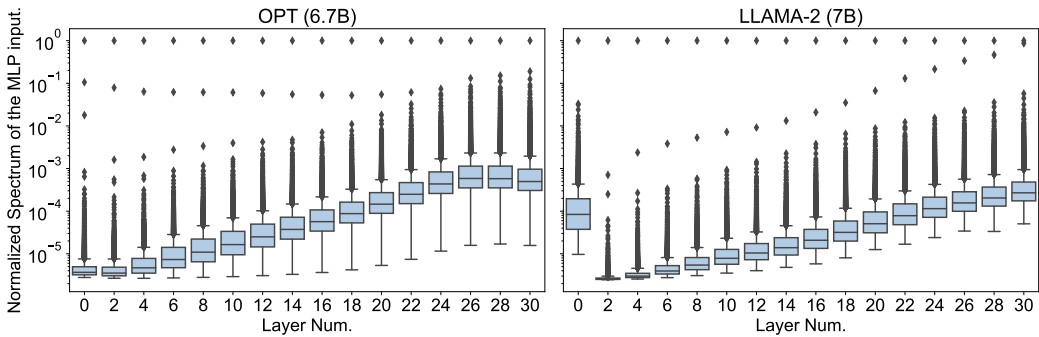

Figure 7: Normalized (by maximum) spectrum of the MLP inputs (log scale) using 64 samples. Except for the first layer in the LLAMA-2 model, the eigenvalue distributions for both models show faster decay in early layers compared to later ones. This suggests that a greater amount of slicing could be applied after the orthogonal transformation in these early layers.

We can use the above insights to slice different layers by different amounts. Instead of specifying the slicing level upfront, we set the fraction of the total variance to discard during each PCA calculation, which sets the number of rows and columns to slice off from each matrix. For each model, we run three experiments with varying target variances to obtain a total reduction on the network close to 25%.

The results are shown in Table 5 below. Varying the slicing level by layer improves the WikiText-2 perplexity in OPT models, but has the opposite effect in LLAMA-2 models.

Table 5: Evaluating the effects of varying slicing level by layer. The calibration set size is 128 and the sequence length is the maximum for each model.

| Model | WikiText-2 PPL (25% constant slicing) | WikiText-2 PPL (varying slicing by layer) | Improvement |
|---|---|---|---|
| OPT 6.7B | 12.10 | 11.94, 24.7% total slicing | 0.16 |
| OPT 13B | 11.04 | 10.76, 24.2% total slicing | 0.28 |
| OPT 30B | 10.13 | 9.95, 24.8% total slicing | 0.18 |
| OPT 66B | 9.75 | 9.63, 24.1% total slicing | 0.12 |
| LLAMA-2 7B | 6.84 | 7.63, 24.1% total slicing | -0.79 |
| LLAMA-2 13B | 6.00 | 6.17, 23.3% total slicing | -0.17 |
| LLAMA-2 70B | 4.44 | 4.63, 25.5% total slicing | -0.19 |

## A.5   ZERO-SHOT ACCURACY ABLATION OVER CALIBRATION DATASET

Figure 8 shows the average scores achieved by the sliced models across the zero-shot tasks. The top row of the plot shows the mean accuracy when WikiText-2 is used for calibration, and the bottom row shows the accuracy when Alpaca is used for calibration. We observe a similar pattern to the generation task in the results: the OPT models are more amenable to compression than the LLAMA-2 models, and the reduction in accuracy is less pronounced in the larger models. Here we also include the Phi-2 model: we see that sliced versions of the Phi-2 model are comparable with sliced versions of the LLAMA-2 7B model. The largest OPT and LLAMA-2 models can be compressed very effectively, with just a few percentage points loss when removing 30% of the 66B OPT model.

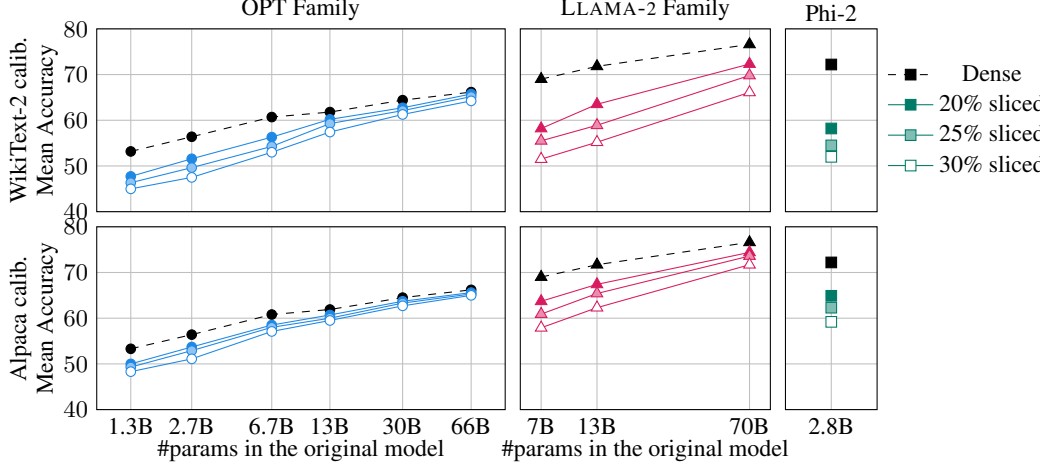

Figure 8: Mean zero-shot accuracy on OPT, LLAMA-2 and Phi-2 across multiple tasks after slicing with the WikiText-2 (top) and Alpaca (bottom) datasets for calibration.

Recovery fine-tuning (RFT) can be applied LLAMA-2 and Phi-2 models to improve their performance further (Figure 5 in main text). Despite an extensive search, we were not able to find RFT parameters that enabled improved performance in the OPT models.

## A.6 DETAILED ZERO-SHOT RESULTS

In this section, we provide the detailed results of the zero-shot tasks we presented in the paper.

Table 6: Downstream zero-shot task performance of OPT, LLAMA-2 and Phi-2 models when slicing using the WikiText2 dataset.

| Model | Slicing | PIQA | WinoGrande | HellaSwag | ARC-e | ARC-c | Avg. Score |
|---|---|---|---|---|---|---|---|
| OPT 1.3B | Dense | 72.42 | 59.27 | 53.72 | 50.97 | 29.52 | 53.18 |
|  | 20% | 65.34 | 54.85 | 45.39 | 46.04 | 26.96 | 47.72 |
|  | 25% | 62.30 | 53.83 | 42.91 | 45.45 | 27.22 | 46.34 |
|  | 30% | 60.77 | 54.70 | 39.81 | 43.90 | 25.77 | 44.99 |
| OPT 2.7B | Dense | 74.81 | 61.01 | 60.58 | 54.42 | 31.14 | 56.39 |
|  | 20% | 68.23 | 57.93 | 51.38 | 51.81 | 28.50 | 51.57 |
|  | 25% | 65.29 | 57.22 | 47.85 | 49.79 | 27.99 | 49.63 |
|  | 30% | 62.35 | 57.22 | 44.18 | 46.72 | 27.05 | 47.50 |
| OPT 6.7B | Dense | 76.39 | 65.19 | 67.16 | 60.14 | 34.64 | 60.70 |
|  | 20% | 72.74 | 61.09 | 61.04 | 55.89 | 30.80 | 56.31 |
|  | 25% | 70.35 | 60.62 | 58.15 | 52.78 | 29.52 | 54.28 |
|  | 30% | 68.61 | 60.69 | 54.56 | 52.15 | 29.01 | 53.00 |
| OPT 13B | Dense | 76.82 | 64.80 | 69.81 | 61.87 | 35.67 | 61.79 |
|  | 20% | 74.48 | 64.96 | 65.42 | 60.90 | 35.24 | 60.20 |
|  | 25% | 73.67 | 64.25 | 63.28 | 60.52 | 34.64 | 59.27 |
|  | 30% | 71.82 | 62.90 | 60.66 | 58.80 | 32.94 | 57.42 |
| OPT 30B | Dense | 78.07 | 68.19 | 72.27 | 65.24 | 38.23 | 64.40 |
|  | 20% | 76.50 | 66.61 | 70.61 | 64.18 | 35.75 | 62.73 |
|  | 25% | 75.30 | 66.61 | 69.42 | 63.55 | 35.67 | 62.11 |
|  | 30% | 74.97 | 65.04 | 68.15 | 63.55 | 34.64 | 61.27 |
| OPT 66B | Dense | 79.82 | 68.90 | 74.85 | 67.21 | 40.02 | 66.16 |
|  | 20% | 78.73 | 67.88 | 73.79 | 68.81 | 39.51 | 65.74 |
|  | 25% | 78.40 | 67.09 | 73.33 | 67.89 | 39.16 | 65.17 |
|  | 30% | 77.42 | 66.30 | 72.62 | 66.90 | 37.97 | 64.24 |
| LLAMA-2 7B | Dense | 79.11 | 69.06 | 75.99 | 74.58 | 46.25 | 69.00 |
|  | 20% | 69.42 | 65.11 | 59.04 | 59.76 | 37.54 | 58.18 |
|  | 25% | 66.87 | 63.38 | 54.16 | 58.46 | 34.56 | 55.48 |
|  | 30% | 63.55 | 61.33 | 49.62 | 51.77 | 31.23 | 51.50 |
| LLAMA-2 13B | Dense | 80.47 | 72.22 | 79.39 | 77.48 | 49.23 | 71.76 |
|  | 20% | 71.87 | 69.38 | 63.04 | 69.87 | 43.09 | 63.45 |
|  | 25% | 68.55 | 67.48 | 58.10 | 62.50 | 37.88 | 58.90 |
|  | 30% | 66.10 | 65.11 | 52.69 | 56.82 | 35.07 | 55.16 |
| LLAMA-2 70B | Dense | 82.70 | 77.98 | 83.84 | 80.98 | 57.34 | 76.57 |
|  | 20% | 76.61 | 76.40 | 72.98 | 80.51 | 55.20 | 72.34 |
|  | 25% | 74.92 | 75.37 | 68.84 | 77.90 | 51.71 | 69.75 |
|  | 30% | 72.31 | 73.56 | 63.69 | 73.40 | 47.61 | 66.11 |
| Phi-2 | Dense | 79.11 | 75.77 | 73.83 | 78.32 | 54.18 | 72.24 |
|  | 20% | 71.87 | 67.80 | 57.76 | 58.00 | 35.32 | 58.15 |
|  | 25% | 69.21 | 65.35 | 52.40 | 53.70 | 31.66 | 54.46 |
|  | 30% | 65.94 | 63.14 | 47.56 | 53.03 | 30.29 | 51.99 |

Table 7: Downstream zero-shot task performance of OPT, LLAMA-2 and Phi-2 models when slicing using the Alpaca dataset.

| Model | Slicing | PIQA | WinoGrande | HellaSwag | ARC-e | ARC-c | Avg. Score |
|---|---|---|---|---|---|---|---|
| OPT 1.3B | Dense | 72.42 | 59.27 | 53.72 | 50.97 | 29.52 | 53.18 |
| | 20% | 69.91 | 55.49 | 47.88 | 49.66 | 27.05 | 50.00 |
| | 25% | 69.37 | 55.72 | 45.82 | 48.70 | 26.62 | 49.25 |
| | 30% | 68.55 | 55.33 | 43.92 | 47.26 | 26.45 | 48.30 |
| OPT 2.7B | Dense | 74.81 | 61.01 | 60.58 | 54.42 | 31.14 | 56.39 |
| | 20% | 71.87 | 58.09 | 54.98 | 54.04 | 29.44 | 53.68 |
| | 25% | 70.95 | 58.09 | 52.62 | 53.03 | 29.61 | 52.86 |
| | 30% | 69.64 | 56.43 | 49.45 | 51.81 | 28.33 | 51.13 |
| OPT 6.7B | Dense | 76.39 | 65.19 | 67.16 | 60.14 | 34.64 | 60.70 |
| | 20% | 74.54 | 62.67 | 62.84 | 59.18 | 33.36 | 58.52 |
| | 25% | 73.78 | 62.59 | 60.99 | 59.01 | 33.70 | 58.01 |
| | 30% | 73.34 | 61.80 | 58.93 | 58.33 | 32.85 | 57.05 |
| OPT 13B | Dense | 76.82 | 64.80 | 69.81 | 61.87 | 35.67 | 61.79 |
| | 20% | 76.01 | 65.19 | 66.15 | 61.57 | 34.73 | 60.73 |
| | 25% | 74.65 | 64.64 | 65.02 | 60.65 | 35.07 | 60.00 |
| | 30% | 74.86 | 63.46 | 63.16 | 61.36 | 34.56 | 59.48 |
| OPT 30B | Dense | 78.07 | 68.19 | 72.27 | 65.24 | 38.23 | 64.40 |
| | 20% | 78.35 | 66.61 | 70.64 | 65.19 | 37.46 | 63.65 |
| | 25% | 77.48 | 65.82 | 69.58 | 65.91 | 37.63 | 63.28 |
| | 30% | 76.93 | 64.96 | 68.66 | 65.70 | 37.12 | 62.67 |
| OPT 66B | Dense | 79.82 | 68.90 | 74.85 | 67.21 | 40.02 | 66.16 |
| | 20% | 79.49 | 68.19 | 73.69 | 67.26 | 39.25 | 65.58 |
| | 25% | 79.11 | 68.35 | 73.30 | 67.00 | 38.74 | 65.30 |
| | 30% | 79.05 | 68.75 | 72.62 | 66.29 | 38.31 | 65.00 |
| LLAMA-2 7B | Dense | 79.11 | 69.06 | 75.99 | 74.58 | 46.25 | 69.00 |
| | 20% | 76.50 | 65.51 | 65.20 | 69.99 | 41.21 | 63.68 |
| | 25% | 74.21 | 64.01 | 60.55 | 66.88 | 38.91 | 60.91 |
| | 30% | 72.25 | 59.83 | 55.86 | 63.93 | 37.80 | 57.93 |
| LLAMA-2 13B | Dense | 80.47 | 72.22 | 79.39 | 77.48 | 49.23 | 71.76 |
| | 20% | 77.97 | 68.90 | 69.64 | 74.71 | 45.99 | 67.44 |
| | 25% | 76.88 | 67.40 | 65.85 | 72.52 | 44.54 | 65.44 |
| | 30% | 74.10 | 65.82 | 60.91 | 68.43 | 42.41 | 62.34 |
| LLAMA-2 70B | Dense | 82.70 | 77.98 | 83.84 | 80.98 | 57.34 | 76.57 |
| | 20% | 81.99 | 76.87 | 78.93 | 80.26 | 54.10 | 74.43 |
| | 25% | 80.69 | 77.98 | 76.97 | 79.67 | 52.65 | 73.59 |
| | 30% | 79.33 | 77.27 | 73.11 | 77.44 | 51.19 | 71.67 |
| Phi-2 | Dense | 79.11 | 75.77 | 73.83 | 78.32 | 54.18 | 72.24 |
| | 20% | 76.17 | 68.75 | 61.95 | 72.18 | 45.48 | 64.90 |
| | 25% | 75.68 | 64.88 | 58.19 | 70.41 | 43.43 | 62.52 |
| | 30% | 74.05 | 62.12 | 53.31 | 67.26 | 39.42 | 63.47 |

Table 8: Downstream zero-shot task performance of LLAMA-2 and Phi-2 models when slicing and recovery fine-tuning using the WikiText2 dataset.

| Model | Slicing | PIQA | WinoGrande | HellaSwag | ARC-e | ARC-c | Avg. Score |
|-------|---------|------|------------|-----------|-------|-------|------------|
| LLAMA-2 7B | Dense | 79.11 | 69.06 | 75.99 | 74.58 | 46.25 | 69.00 |
| | 20% | 69.86 | 64.72 | 61.07 | 54.25 | 36.43 | 57.27 |
| | 25% | 69.26 | 64.96 | 58.65 | 52.36 | 35.75 | 56.20 |
| | 30% | 67.41 | 63.22 | 55.65 | 50.76 | 34.13 | 54.23 |
| LLAMA-2 13B | Dense | 80.47 | 72.22 | 79.39 | 77.48 | 49.23 | 71.76 |
| | 20% | 74.10 | 68.51 | 66.94 | 70.54 | 43.77 | 64.77 |
| | 25% | 71.27 | 68.98 | 64.12 | 63.76 | 40.87 | 61.80 |
| | 30% | 69.64 | 66.85 | 59.93 | 59.55 | 38.65 | 58.93 |
| LLAMA-2 70B | Dense | 82.70 | 77.98 | 83.84 | 80.98 | 57.34 | 76.57 |
| | 20% | 77.86 | 76.16 | 72.91 | 81.27 | 55.89 | 72.82 |
| | 25% | 76.71 | 73.72 | 71.41 | 79.88 | 54.69 | 71.28 |
| | 30% | 75.14 | 73.56 | 69.91 | 74.79 | 51.71 | 69.02 |
| Phi-2 | Dense | 79.11 | 75.77 | 73.83 | 78.32 | 54.18 | 72.24 |
| | 20% | 71.27 | 67.17 | 54.86 | 56.61 | 38.91 | 57.76 |
| | 25% | 69.91 | 65.19 | 52.48 | 52.78 | 35.49 | 55.17 |
| | 30% | 66.16 | 63.54 | 49.72 | 46.38 | 32.68 | 51.70 |

Table 9: Downstream zero-shot task performance of LLAMA-2 and Phi-2 models when slicing and recovery fine-tuning using the Alpaca dataset.

| Model | Slicing | PIQA | WinoGrande | HellaSwag | ARC-e | ARC-c | Avg. Score |
|-------|---------|------|------------|-----------|-------|-------|------------|
| LLAMA-2 7B | Dense | 79.11 | 69.06 | 75.99 | 74.58 | 46.25 | 69.00 |
| | 20% | 76.55 | 65.59 | 68.26 | 71.84 | 45.05 | 65.46 |
| | 25% | 75.79 | 63.22 | 65.12 | 68.22 | 42.83 | 63.04 |
| | 30% | 74.59 | 61.64 | 63.06 | 66.54 | 40.87 | 61.34 |
| LLAMA-2 13B | Dense | 80.47 | 72.22 | 79.39 | 77.48 | 49.23 | 71.76 |
| | 20% | 79.27 | 68.27 | 73.21 | 74.37 | 49.83 | 68.99 |
| | 25% | 78.84 | 67.64 | 71.21 | 73.57 | 49.66 | 68.18 |
| | 30% | 76.11 | 68.03 | 68.58 | 71.42 | 47.10 | 66.35 |
| LLAMA-2 70B | Dense | 82.70 | 77.98 | 83.84 | 80.98 | 57.34 | 76.57 |
| | 20% | 81.94 | 77.74 | 79.39 | 81.57 | 58.45 | 75.82 |
| | 25% | 81.88 | 77.11 | 79.04 | 81.36 | 58.70 | 75.62 |
| | 30% | 80.30 | 75.85 | 77.13 | 80.05 | 58.19 | 74.30 |
| Phi-2 | Dense | 79.11 | 75.77 | 73.83 | 78.32 | 54.18 | 72.24 |
| | 20% | 77.42 | 72.14 | 65.33 | 74.20 | 49.91 | 67.80 |
| | 25% | 76.17 | 68.75 | 63.39 | 70.45 | 47.44 | 65.24 |
| | 30% | 75.24 | 65.59 | 60.10 | 70.16 | 46.25 | 63.47 |

## A.7 BENCHMARKING THROUGHPUT EXPERIMENT

Table 10: Benchmarking throughput for OPT and LLAMA-2 models on 80GB H100 GPUs. We set the sequence length to 128 and find the maximum throughput by doubling the batch size until the GPUs run out of memory or the throughput drops off.

| Model | Slicing | GPUs | Batchsize | Tokens/s |
|---|---|---|---|---|
| | Dense | 1 | 512 | 2518 |
| OPT 13B | 25% | 1 | 512 | 2846 (1.13×) |
| | 50% | 1 | 512 | 3071 (1.22×) |
| | Dense | 2 | 16 | 141 |
| OPT 66B | 25% | 2 | 16 | 152 (1.08×) |
| | 50% | 1 | 32 | 441 (6.26×) |
| | Dense | 1 | 512 | 2707 |
| LLAMA-2 13B | 25% | 1 | 512 | 2878 (1.06×) |
| | 50% | 1 | 512 | 3122 (1.15×) |
| | Dense | 2 | 128 | 541 |
| LLAMA-2 70B | 25% | 2 | 256 | 839 (1.55×) |
| | 50% | 1 | 128 | 1014 (3.75×) |

## A.8 BENCHMARKING INFERENCE TIME OF SLICEGPT AGAINST SPARSEGPT

We use the CuSparseLT 0.5 library to run sparse matrix multiplications on an 80 GB A100 GPU, replicating the size of the matrix-matrix multiplications in three different-sized LLAMA-2 models. We used PyTorch to run similar matrix multiplications for the dense equivalent, and for SliceGPT (which is also straightforward dense matmul, but smaller). We chose a sequence length of 2048, and took the matrix sizes from the HuggingFace config files. We took the median runtime over $10^3$ attempts.

Each LLAMA-2 layer requires a gated FFN with one up projection, one down projection, and a gated projection. In attention, the architecture of the model means that the query matrix multiplication is a different size to the key and value matrix multiplications. The following table shows the time taken in ms to run each matrix multiplication in the model, plus a "total" time and a relative speedup.

Table 11: Results of timing the matrix multiplications involved in each layer of LLAMA-2 models. For larger models, SliceGPT (25%) gives the same speedup as SparseGPT 2:4 but with better WikiText-2 perplexity. For smaller models SparseGPT 2:4 provides better speedup albeit at worse perplexity. Slicing at 50% trades off perplexity for even greater speedups.

| Model | Method | PPL | Operation (ms) | | | | | Total in ms (speedup) |
|---|---|---|---|---|---|---|---|---|
| | | | Down Proj | Up/Gate Proj | K,V | Q | Out | |
| | Dense | 5.47 | 0.89 | 0.87 | 0.34 | 0.34 | 0.34 | 3.99 |
| LLAMA-2 7B | SparseGPT 2:4 | 8.69 | 0.56 | 0.61 | 0.23 | 0.23 | 0.23 | 2.70 (1.48×) |
| | SliceGPT (25%) | 7.24 | 0.67 | 0.64 | 0.26 | 0.25 | 0.27 | 2.99 (1.33×) |
| | SliceGPT (50%) | 17.17 | 0.46 | 0.44 | 0.18 | 0.18 | 0.18 | 2.06 (1.94×) |
| | Dense | 4.88 | 1.29 | 1.28 | 0.52 | 0.52 | 0.52 | 5.93 |
| LLAMA-2 13B | SparseGPT 2:4 | 7.07 | 0.81 | 0.95 | 0.31 | 0.31 | 0.31 | 3.95 (1.50×) |
| | SliceGPT (25%) | 6.30 | 1.03 | 0.98 | 0.39 | 0.39 | 0.41 | 4.57 (1.30×) |
| | SliceGPT (50%) | 13.71 | 0.68 | 0.67 | 0.26 | 0.27 | 0.30 | 3.11 (1.91×) |
| | Dense | 3.32 | 4.63 | 4.27 | 0.21 | 1.27 | 1.27 | 16.13 |
| LLAMA-2 70B | SparseGPT 2:4 | 4.98 | 2.87 | 3.69 | 0.14 | 0.84 | 0.83 | 12.20 (1.32×) |
| | SliceGPT (25%) | 4.60 | 3.4 | 3.26 | 0.16 | 0.96 | 1.00 | 12.20 (1.32×) |
| | SliceGPT (50%) | 8.86 | 2.28 | 2.34 | 0.15 | 0.69 | 0.68 | 8.63 (1.87×) |

We also benchmarked the OPT architecture in the same way. In this case, the matrix multiplications associated with Key, Value, Query and Out are all the same size, and there are just two matrix multiplications in the MLP section (FC1 and FC2).

Table 12: Results of timing the matrix multiplications involved in each layer of OPT models. For larger models, SliceGPT (25%) gives slightly better speedup than SparseGPT 2:4, and with better WikiText-2 perplexity. For smaller models SparseGPT 2:4 provides better speedup albeit at worse perplexity. Slicing at 50% trades off perplexity for even greater speedups.

| Model | Method | PPL | Operation (ms) | | | Total in ms |
| | | | FC2 | FC1 | K,V,Q,Out | (speedup) |
|---|---|---|---|---|---|---|
| OPT 13B | Dense | 10.12 | 1.89 | 1.89 | 0.52 | 7.75 |
| | SparseGPT 2:4 | 11.80 | 1.18 | 1.50 | 0.31 | 5.42 (1.43×) |
| | SliceGPT (25%) | 10.94 | 1.50 | 1.45 | 0.38 | 5.92 (1.31×) |
| | SliceGPT (50%) | 15.39 | 0.96 | 0.99 | 0.26 | 3.98 (1.95×) |
| OPT 30B | Dense | 9.56 | 10.29 | 1.28 | 0.52 | 5.93 |
| | SparseGPT 2:4 | 10.53 | 0.81 | 0.95 | 0.31 | 3.95 (1.50×) |
| | SliceGPT (25%) | 10.04 | 1.03 | 0.98 | 0.39 | 4.55 (1.30×) |
| | SliceGPT (50%) | 12.47 | 0.68 | 0.67 | 0.26 | 3.06 (1.94×) |
| OPT 66B | Dense | 9.33 | 4.63 | 4.27 | 0.21 | 14.01 |
| | SparseGPT 2:4 | 10.22 | 2.87 | 3.69 | 0.14 | 10.81 (1.30×) |
| | SliceGPT (25%) | 9.68 | 3.40 | 3.26 | 0.16 | 10.56 (1.33×) |
| | SliceGPT (50%) | 11.39 | 2.28 | 2.34 | 0.15 | 7.56 (1.85×) |

## A.9 RECOVERY FINE-TUNING COST

All LLAMA-2 , OPT and Phi-2 models can be sliced on a single GPU in 1 to 3 hours. With recovery fine-tuning we compress all LMs in 1 to 5 hours total, as shown in Table 13.

Table 13: Compute cost of slicing 30% with SliceGPT and performing recovery fine-tuning using the Alpaca dataset. Here we use a calibration set size of 1024 for LLAMA-2 models and 2048 for Phi-2 , and calibration sequence length 2048 in all cases.

| Model | SliceGPT 30% | | Recovery fine-tuning | | Total |
| | Time | GPUs | Time | GPUs | |
|---|---|---|---|---|---|
| LLAMA-2 7B | 0h44m | 1xH100 80GB | 0h23m | 1xH100 80GB | 1h07m |
| LLAMA-2 13B | 1h08m | 1xH100 80GB | 0h44m | 1xH100 80GB | 1h52m |
| LLAMA-2 70B | 3h31m | 1xH100 80GB | 1h35m | 4xH100 80GB | 5h06m |
| Phi-2 | 0h49m | 1xV100 32GB | 1h59m | 1xV100 32GB | 2h48m |

