# OpenReview forum: "SliceGPT: Compress Large Language Models by Deleting Rows and Columns"
_ICLR.cc/2024/Conference — ICLR 2024 poster_

### Official Review · Reviewer_jAXk · 2023-10-27

**Soundness:** 3 good
**Presentation:** 3 good
**Contribution:** 3 good
**Rating:** 6
**Confidence:** 4

**Summary:**

The authors of this paper describe their methodology as a transformation of a Transformer network from LayerNorm to RMSNorm. They implement an approach involving the application of orthogonal-matrix transformations and the selective removal of columns and rows from the transformed weight matrices. This process is aimed at reducing the overall model size while preserving performance integrity. The results of their research demonstrate a significant improvement in perplexity on benchmark datasets, OPT and Llmas2, aligning with the 2:4 scheme and underscoring the substantial enhancement in model efficiency and accuracy.

**Strengths:**

* The formulation is clear and enhanced by illuminating diagrams for better comprehension.
* The experimental results illustrate the method's effectiveness, establishing a well-defined trade-off between accuracy and sparsity.

**Weaknesses:**

The experimental section has certain shortcomings:

1. The experiment section does not comprehensively address the comparison between SliceGPT and SparseGPT. While 2:4 sparsity implies a 50% compression rate, Table 1 exclusively showcases SliceGPT with up to 30% compression. This limitation hinders a clear conclusion regarding the superior performance of SliceGPT over SparseGPT.

2. The absence of inference time data for SparseGPT in the experiments makes it challenging to convincingly demonstrate the superior efficiency of SliceGPT.

3. The paper lacks a comparative analysis with state-of-the-art pruning methods such as low-rank approximation, unstructured sparsity, and block sparsity. The omission of these comparisons limits the paper's ability to establish the competitiveness of SliceGPT within the broader context of pruning techniques.

**Questions:**

* Can you show the performance (perplexity, inference time) of SliceGPT at 50% sparsity?
* Can you show the inference time of SparseGPT in comparsion to SliceGPT under the same experimental setup?

**Details Of Ethics Concerns:**

No concern.

---

> ### Author Response · Authors · 2023-11-13
>
> Hi reviewer jAXk, thanks for taking the time to review our work.
>
> > _experiment section does not comprehensively address the comparison between SliceGPT and SparseGPT_
>
> We are working on this, see main rebuttal.
>
> > _The absence of inference time data for SparseGPT in the experiments makes it challenging to convincingly demonstrate the superior efficiency of SliceGPT. (and Q2)_
>
> This was a challenge for us: the SparseGPT authors do not provide a way to verify and kind of speedup using their method. See for example this [github issue](https://github.com/IST-DASLab/sparsegpt/issues/15).  The issue is that pytorch does not provide off-the-shelf support for structured sparsity. This means that at the moment, it’s not possible to verify end-to-end speedup for SparseGPT. Our work-around is to provide end-to-end results for SliceGPT, and apples-to-apples comparisons with SparseGPT on their terms. Please see main rebuttal for more details and forthcoming experiment.
>
> > _lack of comparative analysis with state-of-the-art pruning methods such as low-rank approximation, unstructured sparsity, and block sparsity._
>
> You’re right – we should add some rationale here.
>
> 1. different sparsities. We compared to SparseGPT 2:4 sparsity because this is the SOTA method for LLMs. SparseGPT also provides unstructured sparsity and other sparsity structures, *but these provide little to no speedup*, which is the motivation the developers of  2:4 sparsity. This is why we chose SparseGPT 2:4 as a baseline, because it’s the strongest competitor. If we’re doing better than this, we’re doing well.
>
> 2. Low rank approximations. We did make reference to low-rank works, which take each weight matrix and replace it with a pair of low-rank matrices. To our knowledge, no one has applied this idea to LLMs, and as we discussed in the text, it seems unlikely that they could improve the performance of LLMs. Of course, the [LoRA method]( https://arxiv.org/abs/2106.09685) has been very influential for fine-tuning, but we’re not aware of a method to use low-rank approximations in improving deployment of LLMs.
>
> 3. Other works. We’ve added reference to [LLM-pruner](https://arxiv.org/pdf/2305.11627.pdf), a modern method for pruning LLMs. LLM-pruner provides less speedup than SliceGPT (see their table 3), requires fine-tuning post sparsity, and does not retain perplexity as well as our SliceGPT method (see their table 1). We’ve also added more context around other pruning methods, see response to reviewer 96nz.
>
> >_Can you show the performance (perplexity, inference time) of SliceGPT at 50% sparsity? (Q1)_
>
> Yes, of course. We ran a small experiment for you just now: on OPT66B with 50% removal, SliceGPT achieves a wikitext-2 ppl of 11.39 (against a dense baseline of 9.34). Fixing the sequence length to 128, we were able to achieve a batchsize of 32 on 1x H100 card, leading to a throughput of 440 tokens/s. For the dense model, we required 2x H100 cards and could only fit a batchsize of 16, leading to 141 tokens/s. That’s a 3.12x speedup, and we only use one card! Please read our main rebuttal to understand the nuance of comparing SliceGPT with SparseGPT, as well as a more comprehensive experiment on the effect of batchsize.

---

> > ### Comment · Reviewer_jAXk · 2023-11-15
> >
> > Thank you for the detailed explanation. It's now evident to me that SliceGPT employs compression on both activations and weights, so 25% removal is achieving a similar speedup with a 2:4 sparsity on weights. I can observe that SliceGPT is able to perform comparably to Sparse GPT (2:4) in terms of both speedup and perplexity.
> >
> > A significant contribution appears to be SliceGPT's flexibility, allowing for adjustable slicing levels in both activations and weights, and its compatibility with existing PyTorch operations.
> >
> > The authors have adequately addressed my concerns and provide more comprehensive analysis. I have adjusted my score.

---

> > > ### Author Response · Authors · 2023-11-15
> > >
> > > Hi Reviewer jAXk, thanks for increasing your score. We're glad we were able to address your concerns.

---

### Official Review · Reviewer_fXjP · 2023-10-30

**Soundness:** 4 excellent
**Presentation:** 3 good
**Contribution:** 3 good
**Rating:** 6
**Confidence:** 4

**Summary:**

This paper introduces SliceGPT, a method to reduce the size of matrices for inference of LLMs. The method uses orthogonal matrices to project to a lower-dimensional space the weight matrices, these orthogonal matrices being constructed using PCA.

**Strengths:**

- The paper is well written and pleasant to follow. Ideas are simply explained and figures are helping the understanding.
- Experimental results are convincing.
- I think this method could be really used in practice to reduce inference time.

**Weaknesses:**

- I think section "layernorm transformers can be converted to RMSnorm" is not well motivated. Could the authors explain more in details the subtleties of this section and why it was written? I may have missed the point.
- I'll wait for other reviewers weaknesses to see whether I agree with them.

**Questions:**

- Do the authors plan to release the code? I think open sourcing it is very important for the community.
- The latex is broken, citations are not redirecting, I think your should recompile the pdf.
- Could the authors comment on the use of a random projection (which is orthogonal in expectation, as in sketching methods) compared to $Q_\ell$ computed using by PCA, which is more expensive?
- In practice, not all layers may be equivalent signal-wise, could the authors comment on the possible use of weight watcher ( https://github.com/CalculatedContent/WeightWatcher ) to analyze how to select a different projection dimension for each layer? This question is purely curiosity but I think, combining both SliceGPT and weight watcher could greatly improve the method.
- p8: what do the authors mean by "using dense kernels in our compressed models"? Did they code specific kernels for SliceGPT?
- I think the authors should write a small proof of Equation (2) to increase the readability of the paper. Can the authors provide it in their answer?
- "Theorem" is too strong for Theorem 1, I suggest "Proposition" or "Remark".
- p4: typo: OBC instead of OBS.

Overall I liked the paper and the method, and satisfying answers to my questions and weaknesses would make me consider increase my score.

---

> ### Author Response · Authors · 2023-11-13
>
> > _layernorm transformers can be converted to RMSnorm" is not well motivated._
>
> Apologies if this wasn’t clear. SliceGPT depends on the computational-invariance trick. In turn, this depends on the commutation property of equation (2) – you can apply an orthogonal transform before-and-after RMSNorm without changing the model output. The trick does not work for LayerNorm, so we have to convert a network to RMSNorm before we apply SliceGPT.
>
> > _Do the authors plan to release the code?_
>
> 100%, we feel the same as you. It’s difficult for us to release pre-publication without de-anonymizing ourselves, but our code will be on github with an MIT license as soon as we’re published. If you would like to inspect the code this week, we’re happy to arrange that (though it will take a while for us to go through the code, to be certain that we’re not de-anonymizing ourselves).
>
> > _recompile the pdf_
>
> Will do! We blame overleaf 😊
>
> > _use of a random projection_
>
> We could indeed use a random projection, but it would break eq. 2. It’s not clear to us that random projections could help here, but we hope that you experiment with our code and let us know on release!
>
> > _not all layers may be equivalent signal-wise_
>
> You’re right! We’re preparing such an experiment, see main rebuttal.
>
> > _what do the authors mean by "using dense kernels in our compressed models"?_
>
> Ah, this is something we haven’t communicated very well. Sparse models, like those produced by SparseGPT, are actually not easy to run. Pytorch does not have  off-the-shelf support (kernels) for 2:4 structured sparsity yet. In fact, those authors decline to provide actually accelerated models in this Github issue: https://github.com/IST-DASLab/sparsegpt/issues/15. We simply meant that the models produced by SliceGPT are easy to run in standard pytorch. We were trying to be diplomatic about their (very cool) contribution, whilst letting you know that our code is easy to run.
>
> > _small proof of Equation (2)_
>
> Sure, we can add one to the appendix.
>
> > _"Theorem" is too strong for Theorem 1, I suggest "Proposition" or "Remark"._
>
> Sure, fixed.
>
> > _Typo_
>
> Fixed, thanks.

---

> ### Author Response · Authors · 2023-11-17
> **Update**
>
> Hi Reviewer fXjP,
>
> We hope we managed to give satisfying answers to your questions in our reply above. One of your questions which we did not fully answer was about varying the dimension of each layer: we added experiment (3) in the main rebuttal which demonstrates the feasibility of this, with performance (perplexity) gains for OPT models but degradation in Llama models. We believe varying sparsity/slicing by layer is an important area of research, and hope that a future work can devise a such a scheme for SliceGPT that works for more models than just OPT.
>
> Should you have any further questions, please let us know. If not, we would be very grateful if you were to consider increasing your score.

---

> > ### Comment · Reviewer_fXjP · 2023-11-18
> > **Thanks for the rebuttal**
> >
> > I would like to thank the authors for the extensive experiments they did in their rebuttal.
> > While I think a more principled way to choose the slicing level layer-wise would greatly improve this paper, I think the authors did some interesting preliminary experiments in that direction. \
> > I am therefore increasing my score from 6 to 7. However, since this year at ICLR, we can give only a 6 or a 8, $\textbf{I am keeping my score at 6, but it should be understood as a 7.}$

---

> > > ### Author Response · Authors · 2023-11-19
> > >
> > > Hi Reviewer fXjP, thanks for increasing your score. We're glad we were able to address your concerns.

---

### Official Review · Reviewer_96nz · 2023-10-31

**Soundness:** 3 good
**Presentation:** 4 excellent
**Contribution:** 3 good
**Rating:** 6
**Confidence:** 4

**Summary:**

The authors propose a technique for pruning neurons in Tranformer architectures based on a clever application of orthogonal matrices, which enables PCA-based elimination of rows and columns throughout the architecture. The authors evaluate their technique on a range of large language models from the OPT and Llama-2 families and demonstrate improvements in inference runtime on GPUs.

**Strengths:**

The paper was very well written and organized. I found the method easy to understand. The insights that underpin the method (e.g., invariance to repeated application of orthogonal matrices) are clever and I think the PCA-based pruning of rows and columns in weight matrices is nicely grounded relative to other neuron pruning techniques.

**Weaknesses:**

I think there are two main weaknesses in this paper. First, the authors don’t acknowledge prior work on neuron pruning. Admittedly most of the papers that I’m aware of on this topic focus on convolutional neuron networks. But, some of the methods are likely to provide a reasonable baseline for the proposed technique. I’ve cited some potentially relevant papers below [1, 2, 3, 4, 5].

Second, the results in Table 1 suggest to me that 2:4 sparsity is preferable to the proposed technique? If I understand correctly, 2:4 will remove 50% of the weights in the model and the results in Table 1 show that it suffers less quality degradation than removing 30% of the parameters with SliceGPT. Based on this, I expect 2:4 sparsity would show larger inference runtime savings for a given quality than the results in Table 2.

[1] https://arxiv.org/abs/1708.06519

[2] https://arxiv.org/abs/1707.06342

[3] https://arxiv.org/abs/1707.01213

[4] https://arxiv.org/abs/1707.06168

[5] https://arxiv.org/abs/1810.05270

**Questions:**

I have no additional question.

---

> ### Author Response · Authors · 2023-11-13
>
> Hi Reviewer 96nz, thanks so much for your comments. Here are our replies to your queries.
>
> > _(1) don’t acknowledge prior work on neuron pruning_
>
> You’re right, this is our bad, and we’ve added a new section to the text including those references and more. One thing to note is that LLMs are _so_ much bigger than convnets, that some previous method don’t apply. For example, LeCun’s original OBS method requires O(N^5) operations to prune an NxN weight matrix (an N^3 matrix inversion for each element you want to prune). Completely infeasible on LLMs, where N might be 12k. Nonetheless, we should have cited more work in the pruning literature, even though we didn’t build on it directly.
>
> > _(2) Second, the results in Table 1 suggest to me that 2:4 sparsity is preferable to the proposed technique?_
>
> See our main rebuttal on comparison with SparseGPT.

---

> ### Author Response · Authors · 2023-11-17
> **Update**
>
> Hi Reviewer 96nz,
>
> You were concerned about the performance of SliceGPT compared with SparseGPT in terms of inference runtime and accuracy. We think our main rebuttal provides a clearer comparison of the two methods: we show in experiment (1) that SliceGPT (25%) is the same speed as SparseGPT (50%), before we take improved data movement (and increased possible batch size) into account. Experiment (2) demonstrates the increase in batch size possible with SliceGPT leads to throughput of up to 6.26x that of the dense model, due to memory savings.
>
> Should you have any further questions, please let us know. If not, we would be very grateful if you were to consider increasing your score.

---

### Official Review · Reviewer_gLsa · 2023-10-31

**Soundness:** 3 good
**Presentation:** 3 good
**Contribution:** 3 good
**Rating:** 5
**Confidence:** 3

**Summary:**

This paper introduces SliceGPT - a new approach for compressing large language models. By deleting rows and columns based on computational invariance, SliceGPT can significantly reduce the computation and memory required for inference while maintaining high accuracy. The evaluation demonstrates that this method is effective for large models such as OPT-66B and Llama-70B.

**Strengths:**

- A novel method of compression based on computational invariance
- No special code is required to run the compressed models and achieve speedup and memory savings
- Works for Llama-70B and OPT-66B
- It is well-written and easy to follow

**Weaknesses:**

- The accuracy loss is not "negligible". With 25% sparsity, the perplexity of Llama-2-70B on WikiText2 increases from 3.32 to 4.89, which is similar to a dense Llama-2-13B. However, a 25% sparse Llama-2-70B has much more parameters than a dense Llama-2-13B.
- The speedup is not impressive.
- Compared to the quantization-based method, there is no advantage.

**Questions:**

1. In Table 2, it is not fair to multiply the number of GPUs by the total latency and get "GPUms". Huggingface Transformers implements naive model parallelism (or device placement, or pipeline parallelism without pipelining) method to parallelize the models, which means that only one GPU is active at a time. A correct implementation of tensor parallelism or pipeline parallelism will give different results. Considering this, the latency speedup is less impressive.
2. Give the same parameter count budget or inference latency budget, how does this method compare to quantization-based method?
3. The "computational invariance" trick is similar to a trick in SmoothQuant[1] (equation 3). Both of them multiplicate some matrices between the X and W, so it is good to do some comparison here.

[1] SmoothQuant: Accurate and Efficient Post-Training Quantization for Large Language Models.

---

> ### Author Response · Authors · 2023-11-13
>
> Hi Reviewer gLsa, thanks so much for your comments. Here are our replies to your queries.
>
> > _The accuracy loss is not neglibible_
>
> We agree that this isn’t the right word – we’ve changed the text. We like to point out though that the accuracy loss is less than for SparseGPT (and we have an apples-to-apples latency comparison coming in the next day or so).
>
> > _The speedup is not impressive._
>
> We respectfully disagree! For sparsity/pruning methods, our speedup is pretty good. For example, see the recent LLM-pruner paper. (2305.11627.pdf (arxiv.org) At 20% pruning of LLama 7b, their wikitext ppl is 17.39 at best (up from a baseline of12.62, see their Table 1), and the speedup is less than 15% (see their Table 3). We’re also adding experiments on batch-size (see main rebuttal), where we expect to see much larger throughput increases.
>
> > _Compared to the quantization-based method, there is no advantage (and Q2)_
>
> We agree that quantization is absolutely key for deploying LLMs, and contributes a huge speedup. But pruning and quantization can work together to make LLM deployment cheaper and faster.  Most pipelines use pruning and quantization, along with finetuning/continued pre-training. See for example [Olive](https://microsoft.github.io/Olive/).
>
> There is nothing to prevent a user from applying quantization on top of SliceGPT, and we anticipate that this will give the best speedup. We’d like to emphasize a key point of SliceGPT: that the activations are smaller, leading to less data movement on/between devices, which is something that quantization cannot do. The experiment in our main rebuttal will show this clearly.
>
> > _it is not fair to multiply the number of GPUs by the total latency_
>
> You’re right. If we were to deploy using a more sophisticated deployment that allowed continuous batching, we’d see a speedup of by a factor of up-to the number of GPUs used. Since part of our speedup comes from reduced number of GPUs, the speedup will be less impressive. However, the total energy consumed will still be in proportion to the numbers in our text. We’ve updated the text to reflect this nuance, here’s what we added:
>
> _Our HuggingFace-based testing does not enjoy continuous batching, where the next batch can be loaded onto the first GPU whilst the second GPU processes the current batch. This means that in terms of inference time, the dense-model could be improved more than our sliced model in terms of GPUms. Nonetheless, our measurements \emph{do} reflect the energy-usage per token in such a deployment._
>
> > _The "computational invariance" trick is similar to a trick in SmoothQuant_
>
> You’re right, SmoothQuant also modifies the model, before quantizing. We’ve added a reference to this effect, thanks!
> Again, we're not competing with SmoothQuant, or any other quantization method. We fully intend for SliceGPT to be used in pipelines where quantization is also applied.

---

> > ### Author Response · Authors · 2023-11-17
> > **Update**
> >
> > Hi Reviewer gLsa,
> >
> > You were concerned about accuracy loss and speedups. We think our rebuttal provides some better clarity: we show in experiment (1) that 25% slicing has better perplexity and the same speed as 50% sparsity (SparseGPT), before we take improved data movement (and increased possible batch size) into account. Experiment (2) demonstrates the increase in batch size possible with SliceGPT leads to throughput of up to 6.26x that of the dense model, due to memory savings.
> >
> > Should you have any further questions, please let us know. If not, we would be very grateful if you were to consider increasing your score.

---

> ### Comment · Reviewer_gLsa · 2023-11-22
> **Keeping my initial score**
>
> I've read the rebuttal and appreciate the authors' improvements to their experiments. However, they still note key issues like accuracy loss and limited benefits compared to quantization. So, I'll stick with my initial score.

---

> > ### Author Response · Authors · 2023-11-22
> >
> > Hi gLsa,
> >
> > Thanks for getting back to us. We firmly refute your positioning of pruning _vs_ quantization. Pruning and quantization go hand-in-hand: once the model has been pruned, quantization may be applied afterwards. See for example Appendix C2 of [LLM-pruner](https://arxiv.org/pdf/2305.11627.pdf).
> >
> > It is true that SliceGPT leads to a small accuracy loss: we show that 25% slicing has **superior** accuracy to SparseGPT 2:4, in all the models we tested (see Table 1, main text), with similar performance (see main rebuttal).
> >
> > SliceGPT provides real-world throughput increases, using current pytorch code, in float16. We think this makes a significant contribution to the field of compressing language models.

---

### Official Review · Reviewer_7EAi · 2023-11-02

**Soundness:** 3 good
**Presentation:** 3 good
**Contribution:** 2 fair
**Rating:** 5
**Confidence:** 5

**Summary:**

The paper proposes to idea of using computation invariance for row-/column-wise sparsification. The authors leverages the idea of pre- and post-multiplying each block in a transformer model by orthogonal matrices that warrants computational invariance of each block. On the surface, adding new operations increases the raw FLOPs. However, following this technique, the authors show that they can sparsify most of operations in a transformer models, including attention and FFN layers.

**Strengths:**

$\mathtt{+}$ The idea of computational invariance and re-purposing additional computation for higher opportunity for sparsification is interesting and warrants further investigation.

$\mathtt{+}$ The results are promising and show the benefits across a range of SOTA models. The comparison with SparseGPT technique is also valuable.

**Weaknesses:**

$\mathtt{-}$ The paper lacks sufficient insights of how the rows and columns are sparsified. It was not clear whether some operations are friendlier to row vs. column sparsification or this is a byproduct of the computational invariance approach.

$\mathtt{-}$ The paper compares accuracy with 2:4 structured sparsity but does not provide head-to-head comparison with SparseGPT (2:4) in terms of latency.

$\mathtt{-}$ One of the premises of the paper is memory saving, but going through the results it is not clear how the memory savings are in comparison to 2:4 sparsity. Showing a trade-off possibly can clarify this point.

**Questions:**

I think if the authors could clarify the following questions/comments and include few additional results, the quality of the paper could significantly increase:

(Q1) Show latency comparison across different baselines, (a) Dense, (b) SliceGPT, (c) SparseGPT.

(Q2) I may have missed this in the paper, but can you please clarify how you decide on row/column sparsity and how you select them? If the sparsed rows/columns are spread across the matrix, how do you manage to do the multiplication while getting latency benefits? or the overall benefits are derived from memory savings?

(Q3) Do you have any insights as which operation/layer is more sensitive to sparsification? Have you thought of not uniformly sparsifying all the layers? Can looking into the range of values in the weight matrices provide insights on how to apply the sparsificiation (both degree and pattern)?

---

> ### Author Response · Authors · 2023-11-13
>
> Hi Reviewer 7EAi, thanks so much for your comments. Here are some replies to your queries.
>
> > _insights of how the rows and columns are sparsified (Q2)_
>
> We should have made this clear – we delete the rows / columns that correspond to the smallest eigenvalues. By convention, these are the last rows/columns. Since each deletion (D in equation 9) happens between a pair of blocks, the same number of rows must be deleted from an input matrix as columns from the preceding output matrix. Figure 4 shows the deletions as hatched areas.
>
> Since we use the eigenvalues of the covariance matrix of X as the rotation, after applying Q the network remains invariant (as per thm1) but the variance of each column of X will be equal to the eigenvalues. You can see a plot of some example eigenvalue decays in the supplementary material. If the variance of a column is small, we can assume it is zero always: this is the same as deleting the corresponding column of the preceding weight matrix and the corresponding row of the subsequent weight matrix.
>
> >  _does not provide head-to-head comparison with SparseGPT (Q1)_
>
> > _One of the premises of the paper is memory saving…_
>
> > _which operation/layer is more sensitive to sparsification? (Q3)_
>
> We are addressing each of these with a new small experiment, see main rebuttal.

---

> ### Author Response · Authors · 2023-11-17
> **Update**
>
> Hi Reviewer 7EAi,
>
> You were concerned that we hadn’t provided a head-to-head comparison with SparseGPT, and that the memory savings over SparseGPT weren't clear. We think our main rebuttal fixes this: we show in experiment (1) that 25% slicing is the same speed as 50% sparsity, before we take improved data movement (and increased possible batch size) into account. Experiment (2) demonstrates the increase in batch size possible with SliceGPT leads to throughput of up to 6.26x that of the dense model, due to memory savings.
>
> We also demonstrate the feasibility of applying varying slicing by layer - with performance improvements in OPT but the opposite in Llama models, in experiment (3).
>
> Should you have any further questions, please let us know. If not, we would be very grateful if you were to consider increasing your score.

---

### Author Response · Authors · 2023-11-13
**Main Rebuttal**

Hello Reviewers, thank you all so much for taking the time to read and review our paper.

There was a consensus that you enjoyed the paper, and thought it could be impactful. You said: "_results are convincing_"; "_computational invariance idea is interesting_"; "_well written and easy to follow_"; "_no special code required_", "_works for SOTA big models_"; "_this method could be really used in practice to reduce inference time_".

We’re also grateful to you for highlighting some areas where we could have communicated our ideas and method better. In response, we are running three small experiments that we think will alleviate your concerns:

1. **A better comparison of SliceGPT with SparseGPT**.

We failed to communicate something important about the difference between SliceGPT and SparseGPT.  SliceGPT removes entries from the weights _and_ activations, whilst SparseGPT removes them only from the weights. We tried to illustrate this in Figure 1. This means that a 50% reduction with SliceGPT would likely be much faster than 50%-SparseGPT. We will run an experiment to verify this, and post the results here (and update the text) in the next day or two.

Why didn’t we run apples-to-apples comparisons with SparseGPT? Because nobody has actually demonstrated end-to-end speedups with sparse methods in LLMs as yet. In this [github issue](https://github.com/IST-DASLab/sparsegpt/issues/15), the SparseGPT authors say "_SparseGPT itself is just concerned with accurately sparsifying a model; acceleration comes through other software / hardware_". In the appendix of the SparseGPT paper, they provide indications of the best-case speedup for a (huge!) 175B model, by timing the corresponding matmuls. So, our comparison with SparseGPT cannot be end-to-end, but we will do an apples-to-apples comparison on their terms!

What we should have written in the paper is  “SliceGPT gives real-world end-to-end speedups for LLMs, using only existing pytorch operations. This is in contrast to existing methods like SparseGPT, which at the time of writing has not demonstrated end-to-end speedup: this would require adding specialized cuda code to pytorch. This makes SliceGPT the first sparsity method to offer end-to-end speedup for LLMs”.


2. We have not communicated well **the memory-saving benefits of SliceGPT**.

 Because SliceGPT removes part of the signal (activations) as well as weights, it is possible to use larger batches, and data movement on the device (and between devices) is improved. We will add an experiment to quantify this benefit later today.

3. More **insights on where sparsity occurs**.

In the current paper, we applied X% sparsity to each block – we deleted the last N rows or columns of each block such that the sparsity was X% (after rounding). You suggested that we should add insights as to where the sparsity matters, or attempt to vary the sparsity by layer. We will add and experiment to demonstrate the feasibility of this.

We’ll be uploading edits to the paper over the next day or two to include these experiments: we’ll notify you on this forum when we have done so.

Some of you have clarifying questions which are not related to these points, and we will respond to each of your reviews in turn to address these.

---

> ### Author Response · Authors · 2023-11-13
> **(2) Memory saving and batch-size experiment**
>
> We would like to share this experiment to address point (2) in the above: SliceGPT has significant memory benefits since the activations (signal, X) are sliced, as well as the weight matrices.
>
> We ran some tests on 80GB H100 GPUs to compare the token throughput of our models sliced at 25% and 50% compared to the dense model. We set the sequence length to 128, and found the maximum throughput by doubling the batch size until the GPUs ran out of memory or the throughput dropped off. We see that the 25% sliced models achieve up to 1.55x throughput improvement. At 50% slicing we can run the largest models on one GPU instead of two, with large increases in throughput: 3.13x and 1.87x. This means that for a fixed number of GPUs, these models achieve 6.26x and 3.75x throughput of a dense model (last column). We note that the WikiText2 perplexity of SliceGPT at 50% is worse than SparseGPT (yet still not catastrophic!), but the throughput is much higher than could be achieved with a sparse method that does not slice the activations.
>
> For models of size 13B, the performance increase from batch-size increasing is less pronounced because the models take up little of the H100 GPU memory, so we’re already able to get a huge batchsize of 512 with the dense model. On consumer grade GPUs (with less memory) we would expect to see improved throughput for these smaller models too.
>
> Thank you for helping us make this advantage of SliceGPT clearer. We hope this improves your view of SliceGPT: we will be in touch shortly with details on (1) and (3) in the main rebuttal above, as well as a memory/batchsize experiment on smaller devices.
>
>
> | Model | Method | WikiText2 PPL| GPUs | Batchsize | Throughput (token/s) | Thpt. increase (in constant hardware) |
> |-------|---------------|----------------------|------|-----------|----------------------|-----------------------------------|
> |LlaMa2-70B|Dense|3.12|2|128|541||
> |LlaMa2-70B|SliceGPT (25%)|4.35|2|256|839|1.55x|
> |LlaMa2-70B|SliceGPT (50%)|8.86|1|128|1014|3.75x|
> --
> |LlaMa2-13B|Dense|4.58|1|512|2707||
> |LlaMa2-13B|SliceGPT (25%)|5.90|1|512|2878|1.06x|
> |LlaMa2-13B|SliceGPT (50%)|13.71|1|512|3122|1.15x|
> --
> |OPT-66B|Dense|9.32|2|16|141||
> |OPT-66B|SliceGPT (25%)|9.70|2|16|152|1.08x|
> |OPT-66B|SliceGPT (50%)|11.39|1|32|441|6.26x|
> --
> |OPT-13B|Dense|10.11|1|512|2518||
> |OPT-13B|SliceGPT (25%)|10.94|1|512|2846|1.13x|
> |OPT-13B|SliceGPT (50%)|15.39|1|512|3071|1.22x|

---

> ### Author Response · Authors · 2023-11-14
> **(1)	A better comparison of SliceGPT with SparseGPT**
>
> As we discussed above, it is not currently straightforward to run end-to-end evaluations of sparse LLMs. In the SparseGPT paper, those authors provided the relative timing of each operation involved in an OPT transformer layer. This is not a full picture of the relative speedup, yet this is the only way that we can run an apples-to-apples experiment. Above, we demonstrated the advantages of SliceGPT in terms of being able to use bigger batches; here we replicate the experiment in the SparseGPT paper, extended to multiple model sizes, and for Llama-2 architecture.
>
> We used the CuSparseLT 0.5 library to run sparse matmuls on an A100 card, replicating the size of the matrix-matrix multiplications in three different-sized Llama-2 models. We used pytorch to run similar matmuls for the dense equivalent, and for SliceGPT (which is also straightforward dense matmul, but smaller). We chose a sequence length of 2048, and took the matrix sizes from the HuggingFace config files. We took the median runtime over 100 attempts.
>
> Each Llama-2 layer requires a gated FFN with one _up projection_, one _down projection_, and a _gated_ projection. In attention, the architecture of the model means that the query matmul is a different size to the key and value matmuls. The following table shows the time taken in ms to run each matmul in the model, plus a “total” time and a relative speedup.
>
> | Model | Method | WikiText2 PPL | Down Proj (ms)| Up/Gate Proj (ms)| K,V (ms)| Q (ms)| Out (ms)| Total Time (ms) | Relative Speedup |
> |-----------|-----------|--------------|-----------|--------------|------|------|------|-----------------|------------------|
> | LLaMa2-7B | Dense | 5.11 | 0.89 | 0.87 | 0.34 | 0.34 | 0.34 | 3.99 | |
> | LLaMa2-7B | SparseGPT 2:4| 8.15 |0.56 | 0.61 | 0.23 | 0.23 | 0.23 | 2.70 | 1.48x |
> | LLaMa2-7B | SliceGPT (25%)| 6.70 |0.67 | 0.64 | 0.26 | 0.25 | 0.27 | 2.99 | 1.33x |
> | LLaMa2-7B | SliceGPT (50%)| 17.17 |0.46 | 0.44 | 0.18 | 0.18 | 0.18 | 2.06 | 1.94x |
> --
> | LLaMa2-13B| Dense | 4.58 | 1.29 | 1.28 | 0.52 | 0.52 | 0.52 | 5.93 | |
> | LLaMa2-13B| SparseGPT 2:4| 6.63 |0.81 | 0.95 | 0.31 | 0.31 | 0.31 | 3.95 | 1.50x |
> | LLaMa2-13B| SliceGPT (25%)| 5.90 |1.03 | 0.98 | 0.39 | 0.39 | 0.41 | 4.57 | 1.30x |
> | LLaMa2-13B| SliceGPT (50%)| 13.71 |0.68 | 0.67 | 0.26 | 0.27 | 0.30 | 3.11 | 1.91x |
> --
> | LLaMa2-70B| Dense | 3.12 |4.63 | 4.27 | 0.21 | 1.27 | 1.27 | 16.13 | |
> | LLaMa2-70B| SparseGPT 2:4| 4.70 |2.87 | 3.69 | 0.14 | 0.84 | 0.83 | 12.20 | 1.32x |
> | LLaMa2-70B| SliceGPT (25%)| 4.35 |3.4 | 3.26 | 0.16 | 0.96 | 1.00 | 12.20 | 1.32x |
> | LLaMa2-70B| SliceGPT (50%)| 8.86 |2.28 | 2.34 | 0.15 | 0.69 | 0.68 | 8.63 | 1.87x |
>
>
> We see that the SparseGPT 2:4 scheme is far from providing a 2x throughput improvement, even though the weights require half the original storage. For the biggest model, SparseGPT 2:4 gives a 1.32x relative speedup, **exactly the same** as SliceGPT 25%, which has superior perplexity! For the smaller models, the speedup of SparseGPT is a little better than SliceGPT (1.48 vs 1.33). As expected, SliceGPT 50% gives a huge speedup in all cases near 1.9x (though admittedly with worse perplexity).
>
>
> We also benchmarked the OPT architecture in the same way. In this case, the matmuls associated with Key, Value, Query and _out_ are all the same size, and there are just two matmuls in the MLP section (fc1 and fc2).
>
> | Model | Method | WikiText2 PPL | FC2 (ms)| FC1 (ms)| K,V,Q,Out (ms)| Total Time (ms) | Relative Speedup |
> |--|--|--|--|--|--|--|--|
> |OPT-13B|Dense|10.13|1.89|1.89|0.52|7.75||
> |OPT-13B|SparseGPT 2:4|11.80|1.18|1.50|0.31|5.42|1.43x|
> |OPT-13B|SliceGPT (25%)|10.94|1.50|1.45|0.38|5.92|1.31x|
> |OPT-13B|SliceGPT (50%)|15.39|0.96|0.99|0.26|3.98|1.95x|
> --
> |OPT-30B|Dense|9.56|10.29|1.28|0.52|5.93||
> |OPT-30B|SparseGPT 2:4|10.53|0.81|0.95|0.31|3.95|1.50x|
> |OPT-30B|SliceGPT (25%)|10.05|1.03|0.98|0.39|4.55|1.30x|
> |OPT-30B|SliceGPT (50%)|12.47|0.68|0.67|0.26|3.06|1.94x|
> --
> |OPT-66B|Dense|9.34|4.63|4.27|0.21|14.01||
> |OPT-66B|SparseGPT 2:4|10.22|2.87|3.69|0.14|10.81|1.30x|
> |OPT-66B|SliceGPT (25%)|9.70|3.40|3.26|0.16|10.56|1.33x|
> |OPT-66B|SliceGPT (50%)|11.39|2.28|2.34|0.15|7.56|1.85x|
>
> For the OPT case, we see that SliceGPT 25% is actually _slightly faster_ than SparseGPT 2:4 for the largest model size. On the smallest models, the results are closer than for the Llama-2 case (1.43x vs 1.31x).
>
> Overall, the speedup of SliceGPT at 25% is competitive with SparseGPT 2:4. We note that this benchmark is very much on “SparseGPT’s terms” in that data movement is not accounted for, which our above experiment suggests is superior in SliceGPT due to the reduced activation (signal) size.
>
> SliceGPT also has the advantage that you can run it right now, end-to-end, using only existing pytorch code 😊
>
> We hope these experiments clarify the situation regarding relative performance of SliceGPT and SparseGPT.

---

> ### Author Response · Authors · 2023-11-14
> **(3) Varying slicing level by layer experiment**
>
> We would like to share this experiment to address point (3) in the above: varying slicing level by layer.
>
> The reviewers suggested inspecting the impact of varying the level of slicing by block; suggesting that some parts of the network might be more sensitive than others. A natural metric to use for deciding on how much to delete from each block is the eigenvalue decay: Instead of specifying the slicing level upfront, we set the fraction of the total variance (=eigenvalues) to discard during each PCA calculation, which sets the number of rows and columns to slice off from each matrix. For each model, we ran three experiments with varying target variances to obtain a total reduction on the network close to 25%.
>
> The results are shown in the table below: we show the perplexity level with uniform slicing; with the new slicing scheme and the improvement.  Varying the slicing level by layer improves the WikiText2 perplexity in OPT models, but has the opposite effect in Llama-2 models. The schedules themselves can be seen in the _existing_ supplementary, where we have provided boxplots of the eigenvalues.
>
> | Model      | WikiText2 ppl at 25% constant slicing | WikiText2 ppl varying slicing by layer | WikiText2 ppl improvement |
> | ---------- | ----- | ------- | ------- |
> | OPT-6.7B   | 12.10| 11.94, 24.7% total slicing| 0.16 |
> | OPT-13B    | 11.04| 10.76, 24.2% total slicing| 0.28 |
> | OPT-30B    | 10.13| 9.95, 24.8% total slicing| 0.18 |
> | OPT-66B    | 9.75  | 9.63, 24.1% total slicing| 0.12 |
> --
> | LlaMa2-7B  | 6.84 | 7.63, 24.1% total slicing| -0.79 |
> | LlaMa2-13B | 6.00| 6.17, 23.3% total slicing|-0.17 |
> | LlaMa2-70B | 4.44 | 4.63, 25.5% total slicing| -0.19 |
>
> There have been a few previous works on designing a sparsity scheme, including  [OWL]( https://arxiv.org/abs/2310.05175). We’ve added our small experiment to the supplementary material, and we hope that a future work can devise a layerwise slicing scheme for SliceGPT that works for more models than just OPT.
>
> We trust this sates the reviewer’s curiosity regarding layerwise slicing 😊.
>
> We look forward to engaging in further fruitful discussions with the reviewers in light of the above experiments.

---

### Author Response · Authors · 2023-11-15
**Update from the authors**

Dear Reviewers and AC,

Thank you again for your helpful comments and for giving us the opportunity to better communicate the strengths and weaknesses of SliceGPT. We have now responded to each of you individually and have added experiments to address the three points that we identified in the main rebuttal. We would love to know if we addressed your concerns, and how we could further improve.  We are looking forward to talking to you.

---

> ### Author Response · Authors · 2023-11-16
> **Revisions uploaded**
>
> Hello Reviewers,
>
> Thanks for all your comments that have improved our manuscript. We've uploaded a revised pdf, and a new supplementary material that includes the results of experiments that you suggested.
>
> We hope that we've clarified the relative performance of SliceGPT and our baseline SparseGPT in these experiments. In particular, we emphasize that SliceGPT-25% is as performant or better than SparseGPT 2:4 in terms of latency in many cases.
>
> Best wishes,
> SliceGPT authors.

---

### Meta-Review · Area_Chair_xJYn · 2023-12-18

**Metareview:**

This paper proposes a strategy for compressing state-of-the-art LLMs. Specifically, it involves replacing matrices in the network with smaller versions, reducing the embedding size in the process. The method seems to work well: it reduces the size of models of contemporary interest, it does so in a way that actually leads to performance improvements (a weakness present in much of the pruning literature), and it does better at maintaining quality than other techniques.

The reviewers raised many concerns that are typical of every paper in the pruning literature (and are often difficult to address in their entirety): clean comparisons to baseline methods (code is often hard to come by and results are hard to reproduce), comparisons to lots of other baselines and common techniques (there are always more out there), more networks and datasets (there are always more out there as well), and citing relevant work from a vast literature.

While those concerns are present here as in all pruning work, the authors seem to have done a reasonable job addressing them and I don't find these weaknesses overly concerning. The method isn't the be-all and end-all of efficient inference on LLMs, but it's a step forward for a research area that may potentially be important in practice.

All in all, I recommend acceptance, although I acknowledge the concerns of reviewers who gave lower scores. This is one of those cases where the work is a step forward, but only time will tell whether it ends up being a productive direction.

**Justification For Why Not Higher Score:**

The paper is interesting, but it's one of many techniques for more efficient LLM inference and only time will tell whether it stands out from the crowd.

**Justification For Why Not Lower Score:**

It's a solid contribution, both for LLMs and in general, and it's of practical relevance.

---

### Decision · Program_Chairs · 2024-01-16

Accept (poster)